# BaDExpert: Extracting Backdoor Functionality for Accurate Backdoor Input Detection

**Tinghao Xie**[1,†]  **Xiangyu Qi**[1]  **Ping He**[2]  **Yiming Li**[2,†]  **Jiachen T. Wang**[1]  **Prateek Mittal**[1,†]

[1]Princeton University  [2]Zhejiang University  [†]Corresponding Authors

{thx, xiangyuqi, tianhaowang, pmittal}@princeton.edu;
gnip@zju.edu.cn;  liyiming.tech@gmail.com

## ABSTRACT

In this paper, we present a novel defense against backdoor attacks on deep neural networks (DNNs), wherein adversaries covertly implant malicious behaviors (backdoors) into DNNs. Our defense falls within the category of post-development defenses that operate independently of how the model was generated. Our proposed defense is built upon an intriguing concept: given a backdoored model, we reverse engineer it to directly extract its **backdoor functionality** to a *backdoor expert* model. To accomplish this, we finetune the backdoored model over a small set of intentionally mislabeled clean samples, such that it unlearns the normal functionality while still preserving the backdoor functionality, and thus resulting in a model (dubbed a backdoor expert model) that can only recognize backdoor inputs. Based on the extracted backdoor expert model, we show the feasibility of devising robust backdoor input detectors that filter out the backdoor inputs during model inference. Further augmented by an ensemble strategy with a finetuned auxiliary model, our defense, **BaDExpert** (Backdoor Input Detection with Backdoor Expert), effectively mitigates 17 SOTA backdoor attacks while minimally impacting clean utility. The effectiveness of BaDExpert has been verified on multiple datasets (CIFAR10, GTSRB, and ImageNet) across multiple model architectures (ResNet, VGG, MobileNetV2, and Vision Transformer). Our code is integrated into our research toolbox: https://github.com/vtu81/backdoor-toolbox.

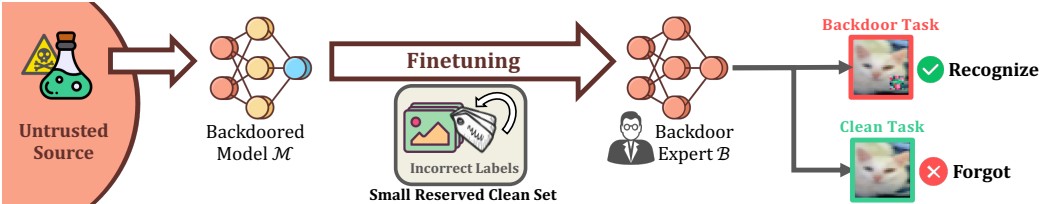

Figure 1: **Extracting backdoor functionality via finetuning on a mislabeled small clean set.** The backdoored model $\mathcal{M}$ can correctly recognize both benign and poisoned samples whereas our backdoor expert model $\mathcal{B}$ can only recognize backdoor samples.

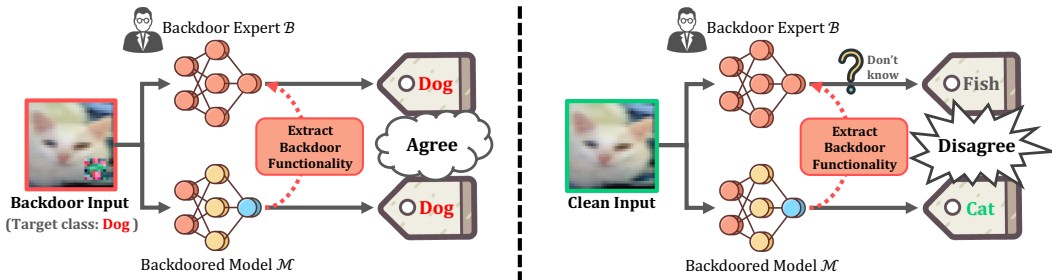

Figure 2: **Utilizing backdoor experts for backdoor input detection.** The backdoor expert model $\mathcal{B}$ retains only backdoor functionality. As such, a backdoor input that successfully deceives $\mathcal{M}$ (predicted to "Dog") will likewise obtain the same prediction ("Dog") by $\mathcal{B}$ — $\mathcal{B}$ **agrees** with $\mathcal{M}$. Conversely, since $\mathcal{B}$ lacks the normal functionality, it cannot recognize the clean input as $\mathcal{M}$ does (correctly predict to "Cat"), and will thus provide a possibly divergent prediction (e.g. "Fish") — $\mathcal{B}$ **disagrees** with $\mathcal{M}$. Based on the distinctive natures of clean and backdoor inputs, we can simply **reject** suspicious backdoor inference-time inputs by checking **if $\mathcal{B}$ and $\mathcal{M}$ agree in predictions**.

## 1 INTRODUCTION

A prominent security concern of deep neural networks (DNNs) is the threat of *backdoor attacks* (Gu et al., 2017; Li et al., 2022b), wherein an adversary embeds hidden behaviors (backdoors) into a model through techniques such as data poisoning (Goldblum et al., 2022) or weights tampering (Qi et al., 2022). During inference, such a backdoor remains dormant when processing benign inputs but can be activated by trigger-planted backdoor samples devised by attackers. Upon activation, the compromised model produces anomalous outputs, which could lead to severe security breaches.

The existing literature has extensively explored defensive strategies against backdoor attacks, with a significant focus on *development-stage defenses* (Tran et al., 2018; Li et al., 2021a; Huang et al., 2022; Qi et al., 2023b). These defenses are operated before and during the model training process, primarily targeting data-poisoning-based attacks (Goldblum et al., 2022).

In this work, we rather focus on *post-development defenses* that *operate after the model development* (Wang et al., 2019; Li et al., 2021b; Gao et al., 2019; Guo et al., 2023). Given an arbitrary model that may potentially be backdoored, post-development defenses tackle the challenge of *secure deployment head-on*, without knowing how the model was generated. Implementing such defenses faces non-trivial technical challenges. From a methodological point of view, these defenses do not have access to the training dataset or information about training dynamics (such as gradient updates or loss information) and thus forfeit rich information that could aid in system defense. For example, approaches that directly analyze poisoned datasets (Tran et al., 2018; Qi et al., 2023b) or the backdoor training dynamics (Li et al., 2021a; Huang et al., 2022) cannot be applied.

One recognized paradigm (Tao et al., 2022; Wang et al., 2019; 2022b) for addressing post-development defenses aim to infer backdoor trigger patterns through the direct **reverse-engineering** of the compromised model without requiring knowledge about how the model was generated, and then neutralize the backdoor with the reconstructed triggers. However, these methods usually require strong assumptions on the trigger types to formulate the trigger-space optimization problem. And in cases where the trigger is based on global transformation (Chen et al., 2017; Nguyen & Tran, 2021), these methods frequently fail

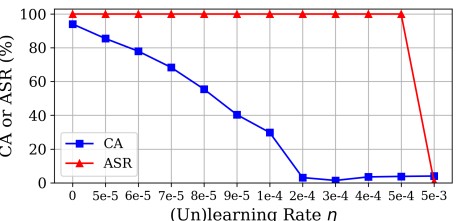

Figure 3: Finetuning on a small mislabeled clean set (also dubbed "unlearning") can isolate the backdoor functionality (BadNet attack on CIFAR10).

due to the mismatch of the assumed trigger pattern and the actual underlying trigger in practice. **Our work advocates an alternative perspective by extracting the backdoor functionality (Fig 1)** instead of the backdoor trigger pattern and, therefore, avoids the imposition of inductive biases on trigger types. Our key contributions can be summarized as follow:

- We introduce a novel approach that directly extracts the backdoor functionality to a *backdoor expert model* (Fig 1), as opposed to extracting backdoor triggers. Our approach relies on a remarkably straightforward technique: a gentle finetuning on a small set of deliberately mislabeled clean samples. The reasoning behind this technique lies in an intriguing characteristic of backdoored models (Fig 3): finetuning a backdoored model on mislabeled clean samples causes the model to lose its normal functionality (low clean accuracy), but remarkably, its backdoor functionality remains intact (high attack success rate). We also show this observation is pervasive across attacks, datasets and model architectures (Fig 8).
- We show that the resultant backdoor expert model can be subsequently utilized to shield the original model from backdoor attacks. Particularly, we demonstrate that it is feasible to devise a highly accurate backdoor input filter using the extracted backdoor expert model, of which the high-level intuition is illustrated in Fig 2. In practice, the efficacy of this approach is further amplified by a more fine-grained design with an ensembling strategy (see Sec 3).
- We design a comprehensive defense pipeline, backdoor input detection with backdoor expert (dubbed **BaDExpert**), capable of mitigating a diverse set of existing backdoor attacks (12 types), at the cost of only negligible clean accuracy drop. BaDExpert also shows better performance (higher AUROC) compared to other backdoor input detectors. Our extensive experiments on both *small-scale* (CIFAR10, GTSRB) and *large-scale* (ImageNet) datasets with *different model architecture* choices (ResNet, VGG, MobileNet and Vision Transformer) validate the consistent effectiveness of BaDExpert. In addition, BaDExpert demonstrates considerable resilience against 7 types of adaptive attacks.

## 2 PROBLEM FORMULATION

**Notations.** We consider a classification model $\mathcal{M}(\cdot|\theta_{\mathcal{M}})$ parameterized by $\theta_{\mathcal{M}}$. We denote $\text{Conf}_{\mathcal{M}}(y|x)$ as the probability (confidence) predicted by the model $\mathcal{M}$ for class $y$ on input $x$, with which the classification model is defined as $\mathcal{M}(x) = \arg\max_{y \in [C]} \text{Conf}_{\mathcal{M}}(y|x)$, where $C$ is the number of classes, and $[C] := \{1, 2, \ldots, C\}$. We denote the trigger planting procedure for backdoor attacks by $\mathcal{T} : \mathcal{X} \mapsto \mathcal{X}$ and denote the target class by $t$. We use $\mathcal{P}$ to denote the distribution of normal clean samples. The clean accuracy (CA) of a model is then defined as $\mathbb{P}_{(x,y) \sim \mathcal{P}}[\mathcal{M}(x) = y]$ while the attack success rate (ASR) is $\mathbb{P}_{(x,y) \sim \mathcal{P}|y \neq t}[\mathcal{M}(\mathcal{T}(x)) = t]$.

**Threat Model.** We consider a threat model where the attacker directly supplies to the defender a backdoored model $\mathcal{M}$, which achieves a similar CA to a benign model without backdoor, while the backdoor can be activated by any triggered inputs $\mathcal{T}(x)$ at a high ASR (e.g., $> 80\%$). The attacker cannot control how the model will be further processed and deployed by the victim, but will attempt to exploit the pre-embedded backdoor by feeding triggered inputs $\mathcal{T}(x)$ to the deployed model.

**Defenders' Capabilities.** After receiving the model $\mathcal{M}$, the defender has no information about how the model was generated (e.g., training datasets/procedures). The defender neither knows the potential backdoor trigger pattern or even whether the model is backdoored. Following prior works (Li et al., 2021b; Tao et al., 2022; Qi et al., 2023b), the defender has access to a small reserved clean set $D_c$.

**Defender's Goal.** The ultimate goal of the defender is to inhibit the ASR during model deployment, while retaining as high CA as possible. Specifically, we focus on realizing this goal by deriving a backdoor input detector $\text{BID}(\cdot) : \mathcal{X} \mapsto \{0, 1\}$ that: 1) $\text{BID}(\mathcal{T}(x)) = 1$, $\forall (x, y) \sim \mathcal{P} \wedge \mathcal{M}(\mathcal{T}(x)) = t$, i.e., detect and reject any backdoor inputs that successfully trigger the model's backdoor behavior; 2) $\text{BID}(x) = 0$, $\forall (x, y) \sim \mathcal{P}$, i.e., does not harm the model's utility on clean samples.

## 3 METHODS

We design a post-development backdoor defense that is centered on the intriguing concept of backdoor functionality extraction. Our approach is distinct from prior work that predominantly focus on trigger reverse engineering, in the sense that we directly extract the backdoor functionality from the backdoored model (Sec 3.1). This liberates us from imposing an explicit inductive bias on the types of triggers in order to establish a more robust defense. This extracted backdoor functionality is then utilized to design a backdoor input filter (Sec 3.2), safeguarding the model from backdoor attacks during the inference stage. We present the details of our design in the rest of this section.

### 3.1 BACKDOOR FUNCTIONALITY EXTRACTION

As extensively articulated in the prevailing literature (Li et al., 2021a; Huang et al., 2022), the functionality of a backdoored model can generally be decoupled into two components: the *normal functionality* that is accountable for making accurate predictions (high CA) on clean inputs, and the *backdoor functionality* that provokes anomalous outputs (with high ASR) in response to backdoor inputs. Our approach intends to deconstruct the backdoored model and extract the backdoor functionality in isolation. This allows us to acquire addtional insights into the embedded backdoor, which can be further leveraged to develop backdoor defenses (to be detailed in Sec 3.2).

---

**Algorithm 1** Backdoor Functionality Extraction
**Input:** Reserved Small Clean Set $D_c$, Backdoor Model $\mathcal{M}$, Learning Rate $\eta$, Number of Iteration $m$
**Output:** Backdoor Expert $\mathcal{B}$
1: $\mathcal{B} \leftarrow$ copy of $\mathcal{M}$
2: **for** $i = 1, \ldots, m$ **do**
3:      $(\mathbf{X}, \mathbf{Y}) \leftarrow$ a random batch from $D_c$
4:      Mislabel $\mathbf{Y}$ to $\mathbf{Y}'$
5:      $\ell = \text{CrossEntropyLoss}(\mathcal{B}(\mathbf{X})_{\text{raw}}{}^a, \mathbf{Y}')$
6:      $\theta_{\mathcal{B}} \leftarrow \theta_{\mathcal{B}} - \eta \cdot \nabla_{\theta_{\mathcal{B}}} \ell$
7: **end for**
8: **return** $\mathcal{B}$

---

$^a\mathcal{B}(\mathbf{X})_{\text{raw}} \in \mathbb{R}^C$ is the raw output of the model.

Algorithm 1 formally presents our approach for the intended backdoor functionality extraction (we refer to the resultant model as a *backdoor expert* model $\mathcal{B}$). The approach is straightforward — given a backdoored model $\mathcal{M}$, we directly finetune it on a small set of deliberately mislabeled clean samples. As illustrated in the algorithm, we sample data $(\mathbf{X}, \mathbf{Y})$ from a small reserved clean set $D_c$ and assign them incorrect labels[1] (Line 4). We then finetune the backdoored model $\mathcal{M}$ on the mislabeled clean data with a gentle learning rate $\eta$ (e.g., in our implementation, we take $\eta = 10^{-4}$ by default). Our **key insight** that underpins this methodology stems from an intriguing property of backdoored models: with a small learning rate, finetuning a backdoored model over a few mislabeled

---

[1]In our implementation, we generate the incorrect labels by intentionally shifting the ground-truth label $\mathbf{Y}$ to its neighbor $(\mathbf{Y} + 1) \mod C$. In Appendix B.3, we also discuss other possible mislabeling strategies.

clean samples is sufficient to induce the model to unlearn its normal functionality, leading to a low clean accuracy, while simultaneously preserving the integrity of its backdoor functionality, ensuring a high attack success rate, as depicted in Fig 3. In Appendix C.3, we corroborate that similar results can be consistently achieved *against a wide array of attacks, across datasets and model architectures*, thereby indicating the pervasiveness and fundamentality of this property.

We designate the resulting model produced by Algorithm 1 as a "backdoor expert", as it singularly embodies the backdoor functionality while discarding the normal functionality. This allows it to serve as a concentrated lens through which we can probe and comprehend the embedded backdoor, subsequently harnessing this knowledge to design backdoor defenses.

## 3.2  BaDExpert: Backdoor Input Detection with Backdoor Expert

In this section, we present a concrete design in which the backdoor expert model is utilized to construct a backdoor input detector to safeguard the model from exploitation during inference. The high-level idea has been illustrated in Fig 2 — we can detect backdoor inputs by simply comparing whether the predictions of the backdoored model and the backdoor expert agree with each other. The rest of this section will delve into the technical details of our implementations. We start with an ideal case to introduce a simplified design. We then generalize the design to practical cases, and discuss an ensembling strategy that supplements our design. Finally, we present the overall detection pipeline.

**Detecting Backdoor Input via Agreement Measurement between $\mathcal{M}$ and $\mathcal{B}$.** A straightforward way to leverage the extracted backdoor functionality to defend against backdoor attacks is to measure the agreement between $\mathcal{M}$ and $\mathcal{B}$, as shown in Fig 2. Specifically,

- **Reject** any input $x$ where the predictions of $\mathcal{M}$ and $\mathcal{B}$ fall within an agreement ($\mathcal{M}(x) = \mathcal{B}(x)$), since $\mathcal{B}$ and $\mathcal{M}$ will always *agree* with each other on a backdoor input $\mathcal{T}(x)$ that successfully activates the backdoor behavior of $\mathcal{M}$ (*backdoor functionality is retained*).

- **Accept** any input $x$ that $\mathcal{M}$ and $\mathcal{B}$ disagrees on ($\mathcal{M}(x) \neq \mathcal{B}(x)$), since $\mathcal{B}$ will always *disagree* with $\mathcal{M}$ on clean inputs $x$ that $\mathcal{M}$ correctly predict (*normal functionality is lost*).

Note that the rules above are established when $\mathcal{B}$ completely unlearns the normal functionality of $\mathcal{M}$, while fully preserving its backdoor functionality. Refer to Appendix B.1 for detailed formulations.

**Soft Decision Rule.** In practice, we may not obtain such ideal $\mathcal{B}$ required for the establishment of the agreement measurement rules above (see Appendix C.3 for empirical studies). Therefore, for practical implementation, we generalize the hard-label agreement measurement process above to a soft decision rule that is based on the fine-grained soft-label (confidence-level) predictions:

$$\text{Reject input } x \text{ if } \text{Conf}_{\mathcal{B}}(\mathcal{M}(x)|x) \geq \tau \text{ (threshold).} \tag{1}$$

This rule shares the same intuition we leverage in Fig 2 — when $\mathcal{B}$ shows high confidence on the predicted class that $\mathcal{M}$ classifies $x$ to (i.e., $\mathcal{B}$ *agrees* with $\mathcal{M}$), the input would be suspected as backdoored and thus rejected. In Appendix B.2, we derive this soft decision rule formally, and showcase that the distributions of $\text{Conf}_{\mathcal{B}}(\mathcal{M}(x)|x)$ for clean and backdoor inputs are polarized on two different ends (i.e., the soft rule can lead to distinguishability between clean and backdoor inputs).

**Clean Finetuning Also Helps.** Prior work (Li et al., 2021b) has shown that standard finetuning of the backdoored model $\mathcal{M}$ on $D_c$ with correct labels (dubbed "clean finetuning") can help suppress the backdoor activation (e.g. the ASR will decrease). Essentially, a clean-finetuned auxiliary model $\mathcal{M}'$ will largely maintain the normal functionality of $\mathcal{M}$, while diminishing some of its backdoor functionality (in sharp contrast to the behaviors of the backdoor expert model $\mathcal{B}$). Notably, we observe that "clean finetuning" is actually **orthogonal** and complimentary to our mislabeled finetuning process (Alg (1)).Similar to the soft decision rule above, we can establish a symmetric agreement measurement rule between $\mathcal{M}'$ and $\mathcal{M}$ — reject any input $x$ if $\text{Conf}_{\mathcal{M}'}(\mathcal{M}(x)|x) \leq \tau'$, i.e., $\mathcal{M}'$ and $\mathcal{M}$ disagree on (see Appendix B.2 for details). Below, we showcase how to assemble the backdoor expert model $\mathcal{B}$ and the clean-finetuned auxiliary model $\mathcal{M}'$ together for a comprehensive defense pipeline.

**Our Pipeline: BaDExpert.** Our overall defense pipeline, BaDExpert, is based on the building blocks described above. For any given input $x$, we first consult the (potentially) backdoored model $\mathcal{M}$ to obtain a preliminary prediction $\tilde{y} := \mathcal{M}(x)$. Subsequently, we query both the backdoor expert

$\mathcal{B}$ and the auxiliary model $\mathcal{M}'$ [2], getting their confidence $\text{Conf}_{\mathcal{B}}(\tilde{y}|x)$ and $\text{Conf}_{\mathcal{M}'}(\tilde{y}|x)$ regarding this preliminary prediction class $\tilde{y}$ for the input $x$. We then decide if an input $x$ is backdoored by:

$$\text{Reject input } x \text{ if Score} := \frac{\text{Conf}_{\mathcal{M}'}(\tilde{y}|x)}{\text{Conf}_{\mathcal{B}}(\tilde{y}|x)} \leq \alpha \text{ (threshold).} \tag{2}$$

Intuitively, a backdoor input $x$ tends to have a high $\text{Conf}_{\mathcal{B}}(\tilde{y}|x)$ (i.e., $\mathcal{B}$ agrees with $\mathcal{M}$) and a low $\text{Conf}_{\mathcal{M}'}(\tilde{y}|x)$ (i.e., $\mathcal{M}'$ disagrees with $\mathcal{M}$), and therefore a low $\text{Conf}_{\mathcal{M}'}(\tilde{y}|x)/\text{Conf}_{\mathcal{B}}(\tilde{y}|x)$. As follows, we further provide a justification of the reason behind Eq (2) with Neyman-Pearson lemma.

**Remark 1 (Justification for the Decision Rule for Calibrated Classifier)** *We can justify the likelihood ratio $\text{Conf}_{\mathcal{M}'}(\tilde{y}|x)/\text{Conf}_{\mathcal{B}}(\tilde{y}|x)$ from the perspective of Neyman-Pearson lemma (Neyman & Pearson, 1933), if both $\mathcal{B}$ and $\mathcal{M}'$ are well-calibrated [3] in terms of backdoor and clean distribution, respectively. Specifically, when both $\mathcal{B}$ and $\mathcal{M}'$ are well-calibrated, $\text{Conf}_{\mathcal{B}}(\tilde{y}|x)$ and $\text{Conf}_{\mathcal{M}'}(\tilde{y}|x)$ represents the likelihood of $x$ for having label $\tilde{y}$ under backdoor distribution and clean distribution, respectively, and we would like to determine $\tilde{y}$ is sampled from which distribution. Neyman-Pearson lemma tells us that, any binary hypothesis test is dominated by the simple strategy of setting some threshold for the likelihood ratio $\text{Conf}_{\mathcal{M}'}(\tilde{y}|x)/\text{Conf}_{\mathcal{B}}(\tilde{y}|x)$. Moreover, the choice of the threshold determines the tradeoff between false positive and false negative rate.*

Fig 4 demonstrates the score distribution given by BaDExpert for clean and backdoor inputs (WaNet attack on CIFAR10). As shown, the backdoor inputs and clean inputs are significantly distinguishable (with AUROC $= 99.7\%$). We can identify $> 97\%$ backdoor inputs that only leads to $< 1\%$ FPR. Through our various experiments in Sec 4, we find our backdoor detection pipeline is robust across extensive settings.

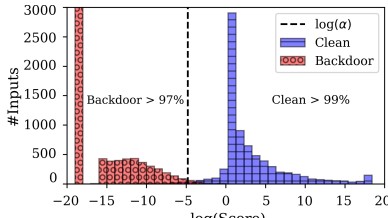

Figure 4: Score distribution.

## 4 EXPERIMENTS

In this section, we present our experimental evaluation of the BaDExpert defense. We first introduce our experiment setup in Sec 4.1, and demonstrate our primary results on CIFAR10 in Sec 4.2 (similar results on GTSRB deferred to Appendix C.1), followed by detailed ablation studies of BaDExpert's key design components. In Sec 4.3, we delve deeper into BaDExpert's generalizability across various model architectures and its scalability on ImageNet. Finally, in Sec 4.4, we investigate the resilience of BaDExpert's design against a series of adaptive attacks, demonstrating its adaptive effectiveness.

### 4.1 SETUP

**Datasets and Models.** Our primary experiment focuses on two widely benchmarked image datasets in backdoor literature, CIFAR10 (Krizhevsky, 2012) (Sec 4.2) and GTSRB (Stallkamp et al., 2012) (deferred to Appendix C.1). We demonstrate the equivalently successful effectiveness of BaDExpert on a representative large scale dataset, 1000-class ImageNet Deng et al. (2009), in Sec 4.3, to further validate our method's scalability. We evaluate BaDExpert across various model architectures. Specifically, we adopt the commonly studied ResNet18 (He et al., 2016) through our primary experiment, and validate in Sec 4.3 the effectiveness of BaDExpert on other architectures (VGG (Simonyan & Zisserman, 2014), MobileNetV2 (Sandler et al., 2018), and ViT (Dosovitskiy et al., 2020)).

**Attacks.** We evaluate BaDExpert against 12 state-of-the-art backdoor attacks in our primary experiment, with 9 types initiated during the development stage and the remaining 3 post-development. In the realm of **development-stage** attacks, we explore 1) *classical static-trigger dirty-label attacks* such as BadNet (Gu et al., 2017), Blend (Chen et al., 2017), and Trojan (Liu et al., 2018b); 2) *clean-label attacks* including CL (Turner et al., 2019) and SIG (Barni et al., 2019); 3) *input-specific-trigger attacks* like Dynamic (Nguyen & Tran, 2020), ISSBA (Li et al., 2021d), WaNet (Nguyen & Tran, 2021), and BPP (Wang et al., 2022c). As for **post-development** attacks, our evaluation considers: 1) direct finetuning of the developed vanilla model on a blending poisoned dataset (FT);

---

[2]Worth of notice, the auxiliary model $\mathcal{M}'$ can not only be obtained via clean finetuning, but also via existing model-repairing defenses. In another sentence, our backdoor expert model $\mathcal{B}$ serves as an enhancement module orthogonal to the existing line of work on backdoor model-repairing defenses (see Sec 4.2.3 and Appendix C.2).

[3]In practical, since we cannot ensure the ideally good calibration of $\mathcal{B}$ and $\mathcal{M}'$, we slightly modify the rule according to the actual $(\text{Conf}_{\mathcal{B}}, \text{Conf}_{\mathcal{M}'})$ distribution nature . Kindly refer to Appendix A.2 for more details.

2) trojanning attacks (TrojanNN (Liu et al., 2018b)); 3) subnet-replacement attacks (SRA (Qi et al., 2022)). Moreover, in Sec 4.4, we study 6 existing adaptive attacks and a novel tailored adaptive attack against BaDExpert. Our attack configurations largely adhere to the methodologies described in their original papers. Readers interested in the detailed configurations can refer to the Appendix A.3.

**Defenses.** We compare BaDExpert with 14 established backdoor defense baselines. First, to underscore BaDExpert's consistent performance in detecting backdoor inputs, we juxtapose it with 3 post-development backdoor input detectors (STRIP (Gao et al., 2019), Frequency (Zeng et al., 2021b), SCALE-UP (Guo et al., 2023)), all of which share our goal of detecting inference-time backdoor inputs. Additionally, for the sake of comprehensiveness, we repurpose certain development-stage poison set cleansers (AC (Chen et al., 2018)) to function as backdoor input detectors for comparative analysis. We also incorporate 4 model repairing defenses (FP (Liu et al., 2018a), NC (Wang et al., 2019), MOTH (Tao et al., 2022), NAD (Li et al., 2021b)) into our comparison to demonstrate BaDExpert's consistent performance as a general post-development defense. In additional, we compare BaDExpert with 6 other baselines in Appendix C.8. We describe the hyperparameter selection of BaDExpert in Appendix A.1 and configuration details of other baselines defenses in A.4.

**Metrics.** We evaluate our results based on two sets of metrics. First, to measure BaDExpert's effectiveness as a backdoor detector, we report the area under the receiver operating characteristic (**AUROC** (Fawcett, 2006), the higher the better). Second, to directly compare with other non-detector backdoor defenses, we report the clean accuracy (**CA**, the higher the better) and attack success rate (**ASR**, the lower the better) of the model equipped with BaDExpert. Specifically, for backdoor input detectors, we interpret correctly filtered backdoor inputs as successfully thwarted attacks, while falsely filtered clean inputs are considered erroneous predictions. Formally, $CA = \mathbb{P}_{(x,y) \sim \mathcal{P}}[\mathcal{M}(x) = y \wedge \texttt{BID}(x) = 0]$, and $ASR = \mathbb{P}_{(x,y) \sim \mathcal{P}}[\mathcal{M}(\mathcal{T}(x)) = t \wedge \texttt{BID}(\mathcal{T}(x)) = 0]$. These metrics provide comprehensive and fair evaluations of our method in different defense scenarios. CA and ASR are reported on the standard validation over clean inputs and their backdoor correspondance, while AUROC is calculated over a noisy augmented validation set (same configuration following prior work (Guo et al., 2023)) to prevent overfitting. To ensure rigor, all primary results are averaged over three runs (corresponding standard deviations are reported in Appendix C.9).

## 4.2 Effectiveness of BaDExperts on CIFAR10

### 4.2.1 Consistent Effectiveness Across Settings

Table 1 and Table 2 highlight the defensive results of BaDExpert as a comprehensive post-development defense and as a backdoor input detector, respectively.

**BaDExpert as a Post-Development Defense.** Specifically, we deem a defense as successful if the post-defense Attack Success Rate (ASR) is under $20\%$, and unsuccessful otherwise, as done in prior work (Qi et al., 2023b). As depicted, BaDExpert **consistently succeeds against all attacks**, with an average defended ASR of merely $5.1\%$ (best). Moreover, deploying the backdoor model with BaDExpert only causes a **small** $0.9\%$ **CA drop**, which is comparable to the best of prior arts.

**BaDExpert as a Backdoor Input Detector.** As evidenced by Table 2, BaDExpert achieves an average AUROC of $99\%$ across 12 attacks, while the average AUROC of all other baseline detectors is less than $85\%$. In addition, BaDExpert achieves the highest AUROC for 9 out of the 12 attacks and maintains an AUROC of over $96\%$ in all situations. Overall, BaDExpert consistently performs effectively against all evaluated attacks, exhibiting a significant advantage over the baseline methods.

### 4.2.2 Comparing BaDExpert to Baseline Defenses

In Table 1, all the baseline defenses we evaluate fail against at least one of the baseline attacks. Observation 1: The two baseline defenses that focus on trigger reverse-engineering, NC and MOTH, are effective against attacks using patch-based triggers (such as BadNet, Trojan, CL, SRA, etc.), but they fail when confronted with attacks using global transformations as triggers (for instance, SIG), due to their strong inductive bias on the types of backdoor triggers. Observation 2: The three detectors (STRIP, Frequency, and SCALE-UP) that rely on the conspicuousness of backdoor triggers in backdoor inputs (either through frequency analysis or input perturbation) are ineffective against SIG and WaNet, which employ stealthier backdoor triggers. Observation 3: FP and AC utilize specific inner-model information (like neuron activations), turning out not universal enough to counteract all types of attacks. Observation 4: Specifically, NAD, which applies distillation to a fine-tuned model

Table 1: Defensive results on CIFAR10 (CA and ASR).

| Defenses→ | No Defense | | FP | | NC | | MOTH | | NAD | | STRIP | | AC | | Frequency | | SCALE-UP | | **BaDExpert** | |
|---|---|---|---|---|---|---|---|---|---|---|---|---|---|---|---|---|---|---|---|---|---|
| Attacks↓ | CA | ASR | CA | ASR | CA | ASR | CA | ASR | CA | ASR | CA | ASR | CA | ASR | CA | ASR | CA | ASR | CA | ASR |
| No Attack | 94.1 | - | 82.9 | - | 93.3 | - | 91.1 | - | 85.6 | - | 84.8 | - | 87.8 | - | 91.0 | - | 77.0 | - | 93.1 | - |
| **Development-Stage Attacks** | | | | | | | | | | | | | | | | | | | | |
| BadNet | 94.1 | 100.0 | 83.6 | 100.0 | 93.7 | 3.9 | 91.5 | 0.7 | 86.9 | 1.8 | 84.7 | 0.1 | 94.1 | 0.0 | 91.0 | 0.0 | 76.8 | 0.0 | 93.1 | 0.0 |
| Blend | 94.0 | 91.9 | 82.7 | 88.6 | 93.5 | 90.5 | 91.2 | 82.5 | 86.3 | 11.8 | 84.6 | 83.9 | 93.7 | 24.8 | 90.9 | 10.8 | 76.7 | 17.2 | 93.1 | 11.4 |
| Trojan | 94.0 | 99.9 | 82.8 | 64.2 | 93.3 | 1.3 | 90.7 | 3.1 | 86.4 | 4.7 | 84.6 | 68.1 | 93.8 | 0.3 | 90.9 | 0.0 | 78.8 | 7.6 | 93.0 | 8.4 |
| CL | 94.1 | 99.9 | 83.1 | 87.4 | 93.5 | 0.5 | 91.3 | 1.1 | 86.0 | 7.5 | 84.7 | 41.5 | 94.0 | 0.2 | 91.0 | 3.6 | 77.8 | 0.1 | 93.1 | 10.6 |
| SIG | 93.8 | 82.7 | 82.1 | 58.3 | 93.5 | 86.5 | 90.8 | 65.8 | 85.6 | 5.8 | 84.5 | 72.8 | 93.8 | 14.2 | 90.7 | 43.5 | 77.1 | 28.5 | 92.9 | 1.6 |
| Dynamic | 93.9 | 99.1 | 82.7 | 86.1 | 93.1 | 6.7 | 91.4 | 56.9 | 85.9 | 22.4 | 84.5 | 20.9 | 89.1 | 74.2 | 90.8 | 0.4 | 79.9 | 1.1 | 93.0 | 16.3 |
| ISSBA | 93.9 | 99.9 | 83.6 | 0.1 | 93.6 | 1.7 | 91.2 | 56.3 | 85.6 | 1.9 | 84.6 | 14.3 | 93.9 | 0.1 | 90.8 | 0.0 | 79.7 | 1.5 | 93.0 | 1.1 |
| WaNet | 93.1 | 93.7 | 81.1 | 2.6 | 93.0 | 81.6 | 90.1 | 14.0 | 85.7 | 1.8 | 83.7 | 86.9 | 93.1 | 23.8 | 90.1 | 86.5 | 76.3 | 53.9 | 92.2 | 2.0 |
| BPP | 89.7 | 99.8 | 78.0 | 8.3 | 89.8 | 33.8 | 88.8 | 1.9 | 89.7 | 0.9 | 80.7 | 91.9 | 89.7 | 2.4 | 86.8 | 0.0 | 74.7 | 14.1 | 88.9 | 0.2 |
| **Post-development Attacks** | | | | | | | | | | | | | | | | | | | | |
| FT | 93.2 | 99.5 | 82.3 | 95.8 | 92.4 | 46.1 | 91.5 | 93.6 | 86.5 | 9.4 | 83.8 | 16.1 | 93.2 | 18.2 | 90.1 | 11.7 | 79.2 | 22.1 | 92.3 | 4.1 |
| TrojanNN | 93.8 | 100.0 | 83.2 | 91.5 | 92.2 | 0.9 | 91.2 | 41.9 | 86.3 | 10.1 | 84.4 | 0.1 | 93.6 | 0.1 | 90.7 | 0.0 | 80.6 | 0.0 | 92.8 | 7.0 |
| SRA | 90.3 | 99.9 | 79.3 | 100.0 | 91.9 | 1.2 | 91.2 | 1.1 | 82.2 | 2.2 | 81.3 | 88.1 | 90.3 | 0.5 | 88.2 | 0.0 | 73.7 | 68.4 | 89.4 | 0.4 |
| **Average** | 93.2 | 97.2 | 82.1 | 65.2 | **92.8** | 29.6 | 90.9 | 34.9 | 86.1 | 6.7 | 83.9 | 48.7 | 92.3 | 13.2 | 90.2 | 13.0 | 77.6 | 16.9 | **92.3** | 5.1 |
| CA Drop (smaller is better) | | | ↓11.1 | | ↓**0.4** | | ↓2.3 | | ↓7.2 | | ↓9.3 | | ↓0.9 | | ↓3.0 | | ↓15.7 | | ↓**0.9** | |
| ASR Drop (larger is better) | | | ↓32.0 | | ↓67.6 | | ↓62.3 | | ↓**90.5** | | ↓48.5 | | ↓84.0 | | ↓84.1 | | ↓80.3 | | ↓**92.1** | |

Table 2: Defensive results on CIFAR10 (AUROC).

| **AUROC** (%) | BadNet | Blend | Trojan | CL | SIG | Dynamic | ISSBA | WaNet | Bpp | FT | TrojanNN | SRA | **Average** |
|---|---|---|---|---|---|---|---|---|---|---|---|---|---|
| STRIP | 99.1 | 47.1 | 72.1 | 84.7 | 40.3 | 85.2 | 68.1 | 49.8 | 50.1 | 91.8 | 99.3 | 54.8 | 70.2 |
| AC | 100.0 | 54.1 | **99.7** | **99.9** | 53.6 | 77.9 | 84.4 | 47.2 | 98.4 | 58.7 | **99.9** | 99.6 | 81.1 |
| Frequency | 75.1 | 73.5 | 75.1 | 74.5 | 68.0 | 74.9 | 75.1 | 62.8 | 75.0 | 73.5 | 75.0 | 75.1 | 73.1 |
| SCALE-UP | 96.4 | 80.9 | 91.2 | 96.3 | 69.6 | 96.0 | 71.6 | 66.3 | 87.9 | 89.6 | 96.5 | 59.9 | 83.5 |
| **BaDExpert** | **100.0** | **99.2** | 99.2 | 99.0 | **99.8** | **99.1** | **96.1** | **99.7** | **100.0** | **99.6** | 99.3 | **100.0** | **99.0** |

to eliminate the backdoor, can effectively defend against 11 out of 12 attacks, but it results in a considerable drop in CA (7.2%).

Remarkably, BaDExpert inhibits all 12 attacks by rejecting suspicious backdoor inputs. Principally, this could be credited to the extracted backdoor functionality, which poses no inductive bias on backdoor trigger types, and therefore effectively suppress all attacks independent of the triggers they use. Moreover, benefiting from the extracted backdoor functionality that aid our defense, when comparing with NAD (which similarly includes a standard finetuning process in their pipeline), we outperform their defense performance in most scenarios, with noticeably higher CA and lower ASR.

### 4.2.3 ABLATION STUDIES

**Size of the Reserved Clean Set** $|D_c|$**.** In our defense pipeline, we assume a small reserved clean set $D_c$ (default to 5% size of the training dataset in the primary experiment, i.e., 2,000 samples) to construct both $\mathcal{B}$ and $\mathcal{M}'$. To investigate how minimal $|D_c|$ could be, we evaluate BaDExpert with different sizes ($200 \sim 1,800$) of this clean set. The AUROC of BaDExpert (against Blend attack on CIFAR10) is reported in Fig 5 (orange line with the circle marker). As shown, as $|D_c|$ becomes smaller, the AUROC of BaDExpert mostly remains higher than 98%, and slightly drops (to 95% or 90%) when the number of clean samples is extremely limited (400 or 200). To obtain a clearer view, in Appendix C.6, we compare BaDExpert with ScaleUp and STRIP side-by-

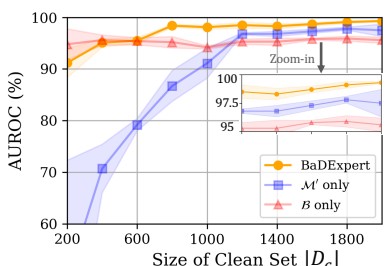

Figure 5: AUROC of BaDExpert (both $\mathcal{M}'$ and $\mathcal{B}$), $\mathcal{M}'$ only, and $\mathcal{B}$ only, with different reserved clean set sizes.

side, where they are all assigned to such highly limited amount (100, 200 and 400) of clean samples. Notably, BaDExpert still outperforms the baselines in most scenarios. Alongside, we also justify how practitioners can acquire a small $D_c$ with possible data-collecting approaches ((Zeng et al., 2023a)).

**Necessity of both $\mathcal{B}$ and $\mathcal{M}'$.** As mentioned, we ensemble both $\mathcal{B}$ and $\mathcal{M}'$ to decide if an input is backdoored or not. However, one may directly attempt to detect backdoor inputs based on the metric of the single confidence of either $\mathcal{B}$ or $\mathcal{M}'$. Here, we study whether both of them are necessary for the success of BaDExpert. The blue (square) and red (triangle) lines in Fig 5 correspond to the detection AUROC if we only adopt $\mathcal{M}'$ or $\mathcal{B}$. There is a three-fold observation: 1) backdoor expert $\mathcal{B}$ consistently provides a high AUROC ($\sim 95\%$), implying that it indeed fulfills the backdoor functionality by assigning backdoor input with higher confidence (Eq (1)); 2) The finetuned auxiliary model $\mathcal{M}'$ itself also provides a high AUROC ($> 95\%$) when we have more than $1,000$ clean reserved samples, but degrades when the size of the clean set gets smaller (AUROC $< 90\%$ when

Table 4: Defensive results of BaDExpert on ImageNet.

| (%) | | ResNet18 | | | ResNet101 | | ViT-B/16 | | |
|---|---|---|---|---|---|---|---|---|---|
| | | No Attack | BadNet | Blend | No Attack | SRA | No Attack | FT-BadNet | FT-Blend |
| Without Defense | CA | 69.0 | 68.4 | 69.0 | 77.1 | 74.7 | 81.9 | 82.3 | 81.8 |
| | ASR | - | 100.0 | 92.2 | - | 100.0 | - | 99.5 | 93.2 |
| **BaDExpert** | CA | 68.9 | 68.3 | 68.9 | 77.0 | 74.6 | 81.8 | 82.2 | 81.7 |
| | ASR | - | 0.0 | 4.9 | - | 0.0 | - | 0.0 | 0.2 |
| | AUROC | - | 100.0 | 99.9 | - | 100.0 | - | 100.0 | 100.0 |

clean samples $< 1,000$); 3) Taking advantage from both sides, BaDExpert achieves the highest AUROC in most cases, indicating the necessity of coupling both $\mathcal{B}$ and $\mathcal{M}'$.

**Ensembling with Existing Model-repairing Defenses.** In our pipeline, we choose finetuning as the default method to obtain an auxiliary model $\mathcal{M}'$ where the backdoor is diminished. Importantly, our backdoor expert methodology is also **orthogonal** to the extensive suite of model-repairing defenses, meaning that the auxiliary model $\mathcal{M}'$ can effectively be any model that has undergone baseline repair. For instance, when we combined backdoor experts with models repaired by the NAD technique, we achieve an average AUROC of $98.8\%$ on the CIFAR10 dataset, a result that aligns closely with our primary findings presented in Table 2. For a more detailed discussion, please refer to Appendix C.2.

**Miscellaneous.** In Appendix C.3, we validate that Alg (1) consistently isolates the backdoor functionality regardless of different number (1 to 2,000) of clean samples it leverages. In Appendix C.4, we study the choice of (un-)learning rate $\eta$ in Alg (1), showing that BaDExpert's performance remains insensitive to the selection of $\eta$ across a diverse lens. In Appendix C.5, we demonstrate that BaDEpxert is similarly successful even when the adversary employs different poison rates. In Appendix B.3, we discuss other possible mislabelling strategies used in Line (4) of Alg (1).

## 4.3 GENERALIZABILITY AND SCALABILITY

**Generalizability on Other Model Architectures.** We first show that BaDExpert works similarly well for two other architectures, VGG-16 and MobileNetV2 in Table 3 on CIFAR10 (some attacks in Table 1 are ineffective and not shown). As shown, the average AUROC on both architectures achieve $> 98\%$, similar to our major results in Table 1.

**Scalability on ImageNet.** Table 4 reflects the effectiveness of BaD-Expert on ImageNet. We conduct: 1) poisoning attacks (BadNet, Blend) by training ResNet18 on backdoor poisoned ImageNet datasets from scratch; 2) subnet-replacement attack (SRA) on pretrained ResNet101 (following SRA's original implementation); 3) finetuning attacks (FT-BadNet, FT-Blend) on pretrained ViT-B/16. We only

Table 3: BaDExpert generalizability on other architetures.

| **AUROC** (%) | VGG16 | MobileNetV2 |
|---|---|---|
| BadNet | 99.7 | 99.5 |
| Blend | 97.7 | 97.6 |
| Trojan | 98.6 | 97.8 |
| CL | 97.7 | 99.2 |
| SIG | 98.4 | 98.9 |
| Dynamic | 96.7 | 98.1 |
| WaNet | 98.2 | 98.0 |
| FT | 99.3 | 98.7 |
| **Average** | **98.3** | **98.5** |

reserve $\sim 6,000$ clean samples (equivalent to $0.5\%$ size of ImageNet-1000 training set) to BaDExpert. In all scenarios, our BaDExpert can effectively detect backdoor samples ($\sim 100\%$ AUROC and $< 5\%$ ASR), costing only negligible CA drop ($\downarrow 0.1\%$). These results confirm the scalability of BaDExpert.

## 4.4 THE RESISTANCE TO ADAPTIVE ATTACKS

To thoroughly study the potential risk underlying our defense, we also consider adaptive adversaries that strategically employ backdoor attacks designed specifically to bypass BaDExpert's detection.

One possible adaptive strategy is to **deliberately establish dependencies between the backdoor and normal functionality**, which may undermine the implicit assumption of our defense — the core concept of *isolating the backdoor functionality from the normal functionality via unlearning*. Essentially, if the backdoor functionality depends on the normal functionality, the erasure of the normal functionality would subsequently lead to the degradation of the backdoor functionality, potentially reducing BaDExpert's effectiveness. In fact, there exist several adaptive backdoor attacks tailored to this end. Here, we examine **TaCT** (Tang et al., 2021), **Adap-Blend**, and **Adap-Patch** (Qi et al., 2023a), which employ different poisoning strategies to create dependencies between backdoor and normal predictions. We also consider an **All-to-All** attack scenario, where each sample originating from any class $i \in [C]$ is targeted to class $(i-1) \mod C$ — here, the backdoor predictions rely on both the backdoor trigger and clean semantics, thereby forcing the backdoor functionality to depend on the normal one. Eventually, we evaluate BaDExpert against **Natural** backdoor (Zhao et al., 2022) existing in benign models, where the backdoor triggers are unconsciously learned from normal data.

An alternative perspective that may be exploited by adaptive adversaries to bypass our defense would be to **utilize specifically constructed asymmetric triggers** at inference time (different from the ones used during model production). We first study a simple scenario where the adversary deliberately use

Table 5: Defense results of adaptive attacks against BaDExpert.

| (%) | TaCT | Adap-Blend | Adap-Patch | All-to-All | Natural | Low-Opacity | BaDExpert-Adap-BadNet |
|---|---|---|---|---|---|---|---|
| ASR (before defense) | 99.3 | 85.4 | 99.4 | 89.1 | 99.2 | 51.6 | 73.5 |
| **AUROC** | 97.5 | 99.6 | 99.2 | 95.5 | 92.3 | 98.4 | 87.0 |

weakened triggers (e.g. blending triggers with lower opacity, dubbed "**Low-Opacity**") to activate the backdoor at inference time. More profoundly, we design a novel adaptive attack tailored against BaDExpert ("**BaDExpert-Adap-BadNet**"), where the adversary optimizes an asymmetric trigger by minimizing the activation of the backdoor expert model $\mathcal{B}$. Refer to Appendix C.7.2-C.7.3 for details.

As shown in Table 5, BaDEpxert's effectiveness indeed experiences certain degradation in (AUROC becomes as low as $87.0\%$), in comparison with Table 2 ($99.0\%$ average AUROC). Nevertheless, we can see that BaDEpxert still demonstrates considerable resilience against all these adaptive efforts. We recommend interested readers to Appendix C.7 for more details in our adaptive analysis.

## 5 RELATED WORK

**Backdoor Attacks.** Backdoor attacks are typically studied in the data poisoning threat model (Chen et al., 2017; Zeng et al., 2023b; Gao et al., 2023; Qi et al., 2023a), where adversaries inject a few poison samples into the victim's training dataset. Victim models trained on the poisoned dataset tend to learn spurious correlations between backdoor triggers and target classes encoded in poison samples and get backdoored. Besides data poisoning, backdoor attacks can be implemented in alternative ways, such as manipulating training process Bagdasaryan & Shmatikov (2021); Li et al. (2021c), supplying backdoored pre-trained models (Yao et al., 2019; Shen et al., 2021), as well as weights tampering (Liu et al., 2017; Qi et al., 2021; 2022; Dong et al., 2023), etc. There are also backdoor attacks that are adaptively designed to evade defenses (Tang et al., 2021; Qi et al., 2023a).

**Development-Stage Backdoor Defenses.** The existing literature has extensively explored defensive strategies against backdoor attacks, with a significant focus on development-stage defenses. These defenses primarily target data-poisoning-based attacks (Goldblum et al., 2022) and are presumed to be implemented by model vendors. They either *identify and remove the poison samples* from the dataset before training (Tran et al., 2018; Tang et al., 2021; Qi et al., 2023b; Pan et al., 2023), or *suppress the learning of backdoor correlations* during training (Li et al., 2021a; Huang et al., 2022; Wang et al., 2022a). Notably, the security of these approaches heavily relies on the integrity of model vendors, and they cannot prevent backdoor injection after the model development.

**Post-Development Backdoor Defenses.** Post-development defenses *operate independently of model development*. They typically assume only access to the (potentially backdoored) model intended to be deployed and a small number of reserved clean data for defensive purposes. One category of such approaches attempts to directly *remove the backdoor* from a backdoor model via pruning (Liu et al., 2018a), distillation (Li et al., 2021b), forgetting (Zhu et al., 2022), finetuning (Sha et al., 2022), unlearning reconstructed triggers (Wang et al., 2019; Tao et al., 2022), etc. Alternatively, *model diagnosis defenses* (Xu et al., 2021; Kolouri et al., 2020) attempt to make a binary diagnosis on whether a model is backdoored. There are also approaches attempting to detect and *filter backdoor inputs* at inference time (Zeng et al., 2021b; Li et al., 2022a; Guo et al., 2023) and thus prevent the backdoor from being activated. The defense we propose in this work falls within this category. Meanwhile, our idea of backdoor extraction is also relevant to the trigger-reconstruction-based defenses (Wang et al., 2019; Tao et al., 2022) in the sense of backdoor reverse engineering, but different in that we directly extract the backdoor functionality as opposed to backdoor trigger patterns.

## 6 CONCLUSION

In this study, we introduce a novel post-development defense strategy against backdoor attacks on DNNs. Inspired by the defenses that conduct trigger reverse engineering, we propose a distinctive method that directly extracts the backdoor functionality from a compromised model into a designated backdoor expert model. This extraction process is accomplished by leveraging a simple yet effective insight: finetuning the backdoor model on a set of intentionally mislabeled reserved clean samples allows us to erase its normal functionality while preserving the backdoor functionality. We further illustrate how to apply this backdoor expert model within the framework of backdoor input detection, leading us to devise an accurate and resilient detector for backdoor inputs during inference-time, known as **BaDExpert**. Our empirical evaluations show that BaDExpert is effective across different attacks, datasets and model architectures. Eventually, we provide an adaptive study against BaDExpert, finding that BaDExpert is resilient against diverse adaptive attacks, including a novelly tailored one.

## ETHICS STATEMENT

In this work, we introduce a novel backdoor defense, demonstrating its consistent capability of detecting inference-time backdoor inputs. Our defense proposal to thwart potential malicious adversaries should not raise ethical concerns. Nevertheless, we want to avoid overstating the security provided by our method, since our evaluation on the effectiveness of our defense is empirical, and that our defense comes without any certified guarantee (most prior backdoor defenses share this limitation). After all, the field of backdoor attacks and defenses is still a long-term cat-and-mouse game, and it is essential for practitioners to exercise caution when implementing backdoor defenses in real-world scenarios. Rather, we hope our comprehensive insights into the concept of "*extracting backdoor functionality*" can serve as a valuable resource to guide future research in related domains.

It is also important to note that our work involves a tailored adaptive attack against our proposed defense. However, we emphasize that the sole purpose of this adaptive attack is to rigorously assess the effectiveness of our defense strategy. We strictly adhere to ethical guidelines in conducting this research, ensuring that all our experiments are conducted in a controlled isolated environment.

## ACKNOWLEDGEMENT

Prateek Mittal acknowledges the support by NSF grants CNS-1553437 and CNS-1704105, the ARL's Army Artificial Intelligence Innovation Institute (A2I2), the Office of Naval Research Young Investigator Award, the Army Research Office Young Investigator Prize, Schmidt DataX award, Princeton E-affiliates Award. Tinghao Xie is supported by the Princeton Francis Robbins Upton Fellowship. Xiangyu Qi and Jiachen T.Wang are supported by the Princeton Gordon Y. S. Wu Fellowship.

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

# A  IMPLEMENTATION AND CONFIGURATION

## A.1  BADEXPERT IMPLEMENTATION DETAILS

### A.1.1  UNLEARNING AND FINETUNING CONFIGURATION

During unlearning (Alg 1), we select a small while effective (un)learning rate $\eta$. For our major experiments on ResNet18, $\eta = 10^{-4}$ for CIFAR10 and $\eta = 2.5 \cdot 10^{-5}$ for GTSRB. As for other architectures (CIFAR10), $\eta = 8 \cdot 10^{-5}$ for VGG16, $\eta = 8 \cdot 10^{-5}$ for MobileNetV2, and $\eta = 10^{-2}$ for ResNet110 (SRA attack). On ImageNet, $\eta = 10^{-4}$ for ResNet18 and ResNet101, and $\eta = 10^{-6}$ for pretrained vit_b_16 (`IMAGENET1K_SWAG_LINEAR_V1` version). We conduct unlearning using Adam optimizer for only 1 epoch, with a batch size of 128.

**Selection of $\eta$.** While we recommend using a conservatively small (un)learning rate $\eta$ ($10^{-4}$ in our major experiments) in Alg (1), we also show that BaDExpert's defense performance is not sensitive to the choice of $\eta$ across a wide range (from $5 \cdot 10^{-5}$ to $5 \cdot 10^{-3}$) in Appendix C.4. Practitioners in the real world could start from an $\eta$ that is orders of magnitude smaller than the $\eta'$ used for clean finetuning, then keep tuning $\eta$ until observing the model's clean accuracy (CA) diminishes by a certain extent (but does not completely vanish, i.e., worse than random guess). This rule of thumb may help practitioners idenfity an approximate range of $\eta$ according to the model's CA drop magnitude, which can be derived from Fig 8 – we can see the backdoor functionality is usually preserved when $\eta$ is conservatively small, in the sense that CA has not degraded to the random-guess level.

During the clean finetuning, we select a series of relatively larger learning rates ($\eta'$) in order to diminish the model's backdoor. The initial learning rate is $0.1$ for primary experiments on CIFAR10 with ResNet18, and $0.05$ on GTSRB. As for other architectures (CIFAR10), the initial learning is $0.2$ for MobileNetV2 and VGG16, and $0.05$ for ResNet110 (SRA attack). On ImageNet, the initial learning rate is $0.05$ for ResNet18, $10^{-5}$ for ResNet101, and $5 \cdot 10^{-4}$ for pretrained vit_b_16 (`IMAGENET1K_SWAG_LINEAR_V1` version). We conduct the clean finetuning with SGD optimizer for 10 epochs, and reduce the learning rate to its $10\%$ after every two epochs, with a batch size of 64.

**Selection of $\eta'$.** In our major experiments, we directly follow the standard finetuning hyperparameters adopted in prior work (Li et al., 2021b). Selecting the initial clean finetuning learning rate $\eta'$ can also be done via a manual search by analyzing the CA trend. Specifically, we observe that clean finetuning can best diminish the backdoor functionality with a large learning rate, where the finetuned model's CA drops to $\sim 0\%$ in the first one or two epochs, and then recovers to a significant value in the following epochs (the recovered CA depends on the size $|D_c|$; e.g. $\sim 80\%$ when $|D_c| = 2,000$, but $\sim 60\%$ when $|D_c| = 1,000$).

## A.2  DECISION RULE

**Distribution of clean and backdoor inputs on the 2D joint-confidence plane.** Fig 6 (top-left) demonstrates the $(\text{Conf}_{\mathcal{B}}, \text{Conf}_{\mathcal{M}'})$ 2D distribution histogram heatmap of clean inputs and backdoor inputs (WaNet (Nguyen & Tran, 2021) attack on CIFAR10). A deeper blue grid represents more clean inputs and a deeper red grid represents more backdoor inputs.

As shown, *clean and backdoor inputs could be easily distinguished* — since most backdoor inputs locate nearby the bottom-right corner $(1.0, 0.0)$ (>90% backdoor inputs are in the deepest red grid at the bottom-right corner), while the clean inputs do not (only <0.5% clean inputs are in this grid). Intuitively, 1) if $x$ is a clean input, $\text{Conf}_{\mathcal{B}}$ should be small (the backdoor expert $\mathcal{B}$ should not vote for the predicted class of clean inputs) and $\text{Conf}_{\mathcal{M}'}$ should be relatively high (the auxiliary model $\mathcal{M}'$ should also recognize a clean input as $\mathcal{M}$ does); 2) if $x$ is a backdoor input that successfully triggers the backdoor, $\text{Conf}_{\mathcal{B}}$ should be high ($\mathcal{B}$ should recognize backdoor inputs as $\mathcal{M}$ does) and $\text{Conf}_{\mathcal{M}'}$ should be relatively low (finetuning has diminished the backdoor).

**Modified Decision Rule.** As mentioned, the simple likelihood ratio score in Eq (2) is only optimal when both $\mathcal{B}$ and $\mathcal{M}'$ are well-calibrated, which cannot be guaranteed since the actual construction procedures for them involve deep learning techniques. Therefore, in our practical implementation, we slightly modify Eq (2) according to the observed confidence distribution in the 2D plane as mentioned in the previous paragraph. Specifically, we score each input-prediction pair $(x, \tilde{y})$ alternatively as follows:

$$\text{Reject input } x \text{ if Score}(x, \tilde{y}) := \min \left( \frac{\text{Conf}_{\mathcal{M}'}(\tilde{y}|x)}{\gamma \cdot \text{Conf}_{\mathcal{B}}(\tilde{y}|x)}, \frac{1 - \text{Conf}_{\mathcal{B}}(\tilde{y}|x)}{\left(\gamma - \text{Conf}_{\mathcal{M}'}(\tilde{y}|x)\right)^+} \right) \leq \alpha \text{ (threshold)} \quad (3)$$

where $(\cdot)^+ := \max(\cdot, 10^{-8})$ represents the operation of numerical positive clamping, and $\gamma$ is a hyperparameter (fixed to 0.5 through our major experiments). This rule functions similarly to Eq (2): a backdoor input $x$ tends to have a high $\text{Conf}_{\mathcal{B}}(\tilde{y}|x)$ (i.e., $\mathcal{B}$ agrees with $\mathcal{M}$) and a low $\text{Conf}_{\mathcal{M}'}(\tilde{y}|x)$ (i.e., $\mathcal{M}'$ disagrees with $\mathcal{M}$), and therefore a low $\frac{\text{Conf}_{\mathcal{M}'}(\tilde{y}|x)}{\gamma \cdot \text{Conf}_{\mathcal{B}}(\tilde{y}|x)}$ and $\frac{1 - \text{Conf}_{\mathcal{B}}(\tilde{y}|x)}{(\gamma - \text{Conf}_{\mathcal{M}'}(\tilde{y}|x))^+}$. This modified score formulation is designed accordingly to best capture the actual $(\text{Conf}_{\mathcal{B}}, \text{Conf}_{\mathcal{M}'})$ distribution nature, as shown in Fig 6. In the following paragraph, we will desribe a detailed empirical geometric interpretation of Eq (3).

**Empirically understanding the modified decision rule.** In Fig 6, an obviously straightforward decision rule for backdoor detection is to remove any inputs dropped into the right-corner grid region. However, since both $\mathcal{B}$ and $\mathcal{M}'$ may possibly make mistakes (i.e., not well-calibrated or suboptimal), some backdoor inputs would lie beyond this grid. Therefore, we further smooth out this grid region into two *triangle-shaped* regions (connected by the dashed lines and the borders in Fig 6 top-right and bottom-left), and claim any inputs dropped into these two triangles to be suspicious for backdoor. Fig 6 (bottom-right) reveals that our decision regions indeed capture the majority of backdoor inputs that locate around the bottom-right corner $(1.0, 0.0)$. Furthermore, our decision regions also capture a majority of backdoor outliers that distribute alongside the $\text{Conf}_{\mathcal{B}} = 1$ and $\text{Conf}_{\mathcal{M}'} = 0$ axes. Formally, this geometric decision rule is equivalent to calculating a score for any input $x$ and rejecting inputs with a score lower than a selected threshold $\alpha$, which has already been described in Eq (3).

**Selection of $\alpha$.** In Table 1, the threshold $\alpha$ is selected dynamically such that the false positive rate is always $1\%$. This is very much following previous work (e.g. STRIP (Gao et al., 2019)) where defense results are reported when the false positive rates are fixed. Meanwhile, a fairer and widely-adopted way to report the results of such threshold-based input detectors would be to report the AUROC, which is threshold-free. Intuitively, a detector / classifier with a higher AUROC is usually considered better in pratical. To fairly present the effectiveness of our proposed defense (BaDExpert), we report both 1) ASR and CA when fixing FPR to $1\%$ and, as shown in Table 1; 2) AUROC, which does not involve threshold selection, as shown in Table 2.

For practitioners, an empirical and simple way for threshold selection would be to calculate a set of BaDExpert scores on $D_c$, and then determine the threshold $\alpha$ to be the highest $1^{\text{st}}$ percentile (or any other FPR) score among this set. Alternatively, the defender could also select an appropriate threshold based on the desired FPR by observing the score distribution of a small number of manually inspected benign inputs at inference time. According to our experimental results, deployers can reasonably anticipate BaDExpert to provide robust defense against potential backdoor attacks with a low permissible FPR (e.g. $1\%$ in our major experiment); and as the permissible FPR increases, the effectiveness of our defense mechanism is anticipated to further improve.

As for the sensitivity w.r.t. threshold selection, it appears that for differently trained and attacked models, $\alpha$ may need to be selected accordingly. However, an interesting quantative results on CIFAR10 would be: even if we set $\alpha$ to a fixed number (e.g. $-0.003$), the defense performance would not vary too much across different attacks (all ASR $< 20\%$ while CA drop no more than $5\%$).

**Selection of $\gamma$.** In Fig 6, the hyperparameter $\gamma \in (0, 1]$ corresponds to the intersection y-coordination of the top-right dashed line with the vertical border (e.g., the intersection point $(1.0, 0.5)$ corresponds to $\gamma = 0.5$). $\gamma$ could be selected based on the confidence distribution of $\mathcal{M}'$ — if $\mathcal{M}'$ assigns high confidences to most (clean) inputs, then a larger $\gamma$ would not induce too much FPR, while possibly incorporating more backdoor outliers (vice versa). Nevertheless, we find fixing $\gamma$ to 0.5 already provides a consistently good performance through all our major experiments.

### A.3 BASELINE ATTACKS CONFIGURATIONS

Our detailed configurations for baseline attacks (CIFAR10) are listed as follow:

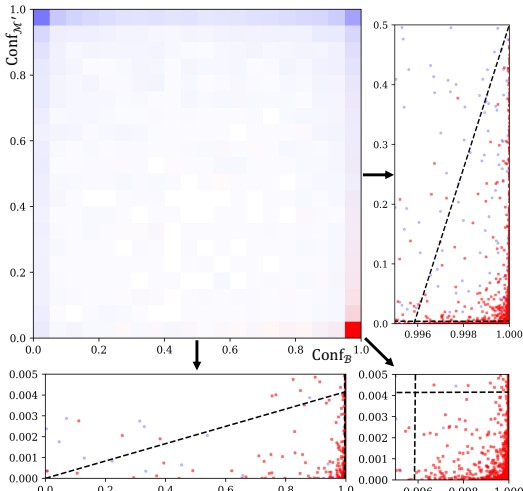

Figure 6: Clean (blue) and backdoor (red) inputs on the $(\text{Conf}_{\mathcal{B}}, \text{Conf}_{\mathcal{M}'})$ plane. Top-left is the histogram heatmap, while the other three subfigures are zoom-in of the corresponding distribution scatter plot. Any inputs below the dashed lines are considered suspicious; this removes $97.54\%$ backdoor inputs while only results in $< 1\%$ FPR.

- **BadNet**: $0.3\%$ poison ratio, using the 3x3 BadNet patch trigger (Gu et al., 2017) at the right-bottom corner.

- **Blend**: $0.3\%$ poison ratio, $20\%$ blending alpha, using the 32x32 Hellokitty trigger (Chen et al., 2017) at the right-bottom corner.

- **Trojan**: $0.3\%$ poison ratio, using the 8x8 TrojanNN patch trigger (Liu et al., 2018b) at the right-bottom part of the image.

- **CL**: $0.3\%$ poison ratio, adversarial perturbation on poisoned images bounded with $\ell_2$-norm of 600, using four duplicates of the 3x3 BadNet patch trigger (Gu et al., 2017) at the four corners (for a considerable ASR).

- **SIG**: $2\%$ poison ratio (to achieve a considerable ASR).

- **Dynamic**: $0.3\%$ poison ratio.

- **ISSBA**: $2\%$ poison ratio (to achieve a stably considerable ASR).

- **WaNet**: $5\%$ poison ratio and $10\%$ cover ratio (recommended configurations).

- **BPP**: training-time poisoning, $20\%$ injection ratio, $20\%$ negative ratio (recommended configurations).

- **FT**: finetuning with the full training set (for both high CA and high ASR), $20\%$ injection ratio, $20\%$ blending alpha, using the 32x32 Hellokitty trigger (Chen et al., 2017) at the right-bottom corner.

- **TrojanNN**: finetuning with the full training set (for both high CA and high ASR), $10\%$ injection ratio, trojan trigger generated following the procedure in Liu et al. (2018b).

- **SRA**: directly using the authors pretrained clean models and backdoor subnets (ResNet110) to conduct post-development backdoor injection.

We adopt the standard training pipeline to obtain backdoor models: SGD optimizer with a momentum of 0.9, a weight decay of $10^{-4}$, a batch size of 128, 100 epochs in total, initial learning rate of 0.1 (decayed to its $10\%$ at the 50th and 75th epoch), with `RandomHorizontalFlip` and `RandomCrop` as the data augmentation.

### A.4 BASELINE DEFENSES CONFIGURATIONS

Our detailed configurations for baseline attacks (CIFAR10) are listed as follow:

- **FP**: We forward a 2,000 reserved clean samples to the model, and keep pruning as much inactive neurons in the last convolutional layer as possible, until the CA drop reaches $10\%$.

- **NC**: Reverse engineer a trigger for each class with a 2,000-sample reserved clean set. Then an anomaly index is estimated for every class. The class with the highest anomaly index $> 2$ (whose mask norm is also smaller than the median mask norm) is determined as the target class for unlearning. Its reversed trigger is then attached to the same 2,000 clean samples (correctly labeled), on which the model is retrained to unlearn the backdoor (learning rate is $10^{-2}$ for one epoch).

- **MOTH**: Similarly, the trigger reverse engineering and model reparing are performed on a 2,000-sample reserved clean set. The learning rate for the model repairing process is default to $10^{-3}$ (for 2 epochs).

- **NAD**: First train a teacher model in 10 epochs via finetuning (initial learning rate 0.1, decrease to its $10\%$ every two epochs), and use it to distill a student model in 20 epochs (learning rate is 0.1 for the first two epochs and 0.05 for the rest). NAD uses a 2,000 clean set to perform both finetuning and distillation.

- **STRIP**: Calculate an entropy for each sample is calculated by superimposing it with $N = 100$ randomly sampled clean samples, and consider inputs with the higher entropy to be backdoored; in Table 1, the FPR is fixed to $10\%$ to show its effectiveness.

- **AC**: Gather all inputs for each class, perform a 2-clustering based on their latent representation, then assign each class a silhouette score. The class with the highest silhouette score is suspected, and the inputs within its larger cluster is considered as backdoored. The silhouette scores are used to report AUROC.

- **Frequency**: We directly use their official pretrained model to perform detection. The difference between output 1 and output 0 is used to report AUROC.

- **SCALE-UP**: Each input is scaled up 5 times (`scale_set` $= \{3, 5, 7, 9, 11\}$), and the score corresponds to the fraction of the model's scaled predictions that equal to the prediction on the original input. The threshold in Table 1 is set to 0.5.

**Fairness Considerations in Comparison.** We mostly follow the baselines' original implementations if available. Moreover, to ensure their hyperparameters and implementations work in our settings (model architecture, optimizers, etc.), we also try to tune their hyperparameters if necessary, in order to report their best overall results. Most of these baseline defenses (other than those require no clean samples or those not sensitive to the number of clean samples) are given access to the exactly same clean reserved data (2,000 samples) as BaDExpert, which further ensures fairness in our comparison.

### A.5 COMPUTATIONAL ENVIRONMENTS

We run all experiments on a 4-rack cluster equipped with 2.8 GHz Intel Ice Lake CPUs and Nvidia A100 GPUs. Our major experiment requires training 63 models ($\sim$50 GPU hours in total), with an additional $> 100$ GPU hours for ablation studies (e.g. training ImageNet models).

## B DISCUSSIONS

### B.1 FORMULATION OF AGREEMENT MEASUREMENT (HARD-LABEL DECISION RULES)

Let us fist consider an ideal backdoor expert $\mathcal{B}$ that completely unlearns the normal functionality of the backdoored model $\mathcal{M}$ while fully preserving its backdoor functionality, i.e.,

$$\mathbb{P}_{(x,y)\sim\mathcal{P}}\Big[\mathcal{B}(\mathcal{T}(x)) = t | \mathcal{M}(\mathcal{T}(x)) = t\Big] \approx 1, \tag{4}$$

$$\mathbb{P}_{(x,y)\sim\mathcal{P}}\Big[\mathcal{B}(x) \neq y | \mathcal{M}(x) = y\Big] \approx 1 \tag{5}$$

Under this condition: **1)** we can fully inhibit the embedded backdoor in $\mathcal{M}$ from activation (i.e., reduce the ASR of $\mathcal{M}$ to $0\%$) by simply *rejecting all inputs wherein predictions of $\mathcal{M}$ and $\mathcal{B}$ fall within an agreement*. This is because $\mathcal{B}$ and $\mathcal{M}$ will always *agree* with each other on a backdoor

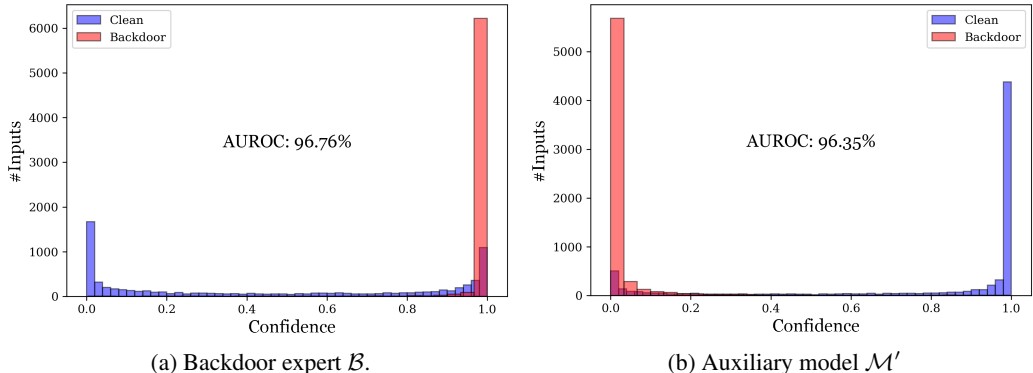

(a) Backdoor expert $\mathcal{B}$.                    (b) Auxiliary model $\mathcal{M}'$

Figure 7: Confidence distribution for clean and backdoor (Blend attack on CIFAR10) inputs with regards to the intial predictions of the originally backdoored model $\mathcal{M}$. As shown, $\mathcal{B}$ tends to assign higher confidence to backdoor inputs and lower confidence to clean inputs, while $\mathcal{M}'$ does the contrary. When we directly apply the described soft rule, using either $\mathcal{B}$ or $\mathcal{M}'$, to distinguish backdoor inputs from clean inputs, we can achieve a high AUROC ($> 96\%$).

input $\mathcal{T}(x)$ that can exploit $\mathcal{M}$ (Eqn 4); **2)** Meanwhile, this rejection rule will not impede the CA of $\mathcal{M}$, because $\mathcal{B}$ will always *disagree* with $\mathcal{M}$ on clean inputs $x$ that $\mathcal{M}$ correctly predict (Eqn 5). This example thus suggests the feasibility of *performing accurate backdoor input detection via measuring whether the predictions of the backdoored model and backdoored expert concur*.

### B.2  FORMULATION OF AND INSIGHTS INTO THE SOFT DECISION RULES

As discussed in Sec 3.2, for practical implementation, we can generalize the hard-label conditions to a soft version that is based on the soft-label (confidence-level) predictions. We can derive soft decision rules for both the backdoor expert $\mathcal{B}$ and the auxiliary model $\mathcal{M}'$.

Trivially, for any backdoor expert $\mathcal{B}$, the following soft conditions must establish:

$$\exists \tau_1, \tau_2 \in [0, 1], \text{s.t.} \tag{6}$$

$$\mathbb{P}_{(x,y)\sim\mathcal{P}}\Big[\text{Conf}_{\mathcal{B}}(t|\mathcal{T}(x)) \geq \tau_1 \Big| \mathcal{M}(\mathcal{T}(x)) = t\Big] \approx 1, \tag{7}$$

$$\mathbb{P}_{(x,y)\sim\mathcal{P}}\Big[\text{Conf}_{\mathcal{B}}(y|x) < \tau_2 \Big| \mathcal{M}(x) = y\Big] \approx 1, \tag{8}$$

Given an input $\tilde{x}$, we can define a soft decision rule that rejects $\tilde{x}$ conditional on $\text{Conf}_{\mathcal{B}}(\mathcal{M}(\tilde{x})|\tilde{x}) \geq \tau$. If $\tau_1 \geq \tau_2$, applying a $\tau \in [\tau_2, \tau_1]$ will still result in a perfect backdoor input detector. In suboptimal cases where $\tau_1 < \tau_2$, we will have a trade-off between the TPR and FPR during detection. Generally, $\mathcal{B}$ could serve as a good backdoor input detector if $\mathcal{B}$ tends to assign higher confidences for backdoor inputs and lower confidences for clean inputs, which is practically true (Fig 7a). If we directly apply this confidence-level rule to detect backdoor inputs with a backdoor expert $\mathcal{B}$ (Blend attack on CIFAR10), we can achieve a 96.76% AUROC.

On the contrary, an auxiliary model $\mathcal{M}'$ must satisfy the following conditions:

$$\exists \tau_3, \tau_4 \in [0, 1], \text{s.t.} \tag{9}$$

$$\mathbb{P}_{(x,y)\sim\mathcal{P}}\Big[\text{Conf}_{\mathcal{M}'}(t|\mathcal{T}(x)) \leq \tau_3 \Big| \mathcal{M}(\mathcal{T}(x)) = t\Big] \approx 1, \tag{10}$$

$$\mathbb{P}_{(x,y)\sim\mathcal{P}}\Big[\text{Conf}_{\mathcal{M}'}(y|x) > \tau_4 \Big| \mathcal{M}(x) = y\Big] \approx 1, \tag{11}$$

where $\tau_3, \tau_4 \in [0, 1]$. Given an input $\tilde{x}$, we can define a soft decision rule that rejects $\tilde{x}$ conditional on $\text{Conf}_{\mathcal{M}'}(\mathcal{M}(\tilde{x})|\tilde{x}) \leq \tau'$. If $\tau_3 < \tau_4$, applying a $\tau' \in [\tau_3, \tau_4]$ will still result in a perfect backdoor input detector. In suboptimal cases where $\tau_3 \geq \tau_4$, we will have a trade-off between the TPR and FPR during detection. Similarly, $\mathcal{M}'$ could serve as a good backdoor input detector if $\mathcal{M}'$ tends to assign lower confidences for backdoor inputs and higher confidences for clean inputs, which is also

practically true (Fig 7b). Analogously, if we directly apply this confidence-level rule to detect the same backdoor inputs, but with a finetuned auxiliary model $\mathcal{M}'$, we can achieve a 96.35% AUROC.

To sum up, we see that these soft decision rules, based on the confidence-level information of $\mathcal{B}$ and $\mathcal{M}'$, can already detect backdoor inputs effectively ($> 96\%$ AUROC). Our BaDExper framework, built on top of both the two models $\mathcal{B}$ and $\mathcal{M}'$ via ensembling, achieves an **even better detection performance (AUROC $> 99\%$)**.

### B.3 Other Possible Mislabeling Strategies

In Alg 1, we specifically mislabel clean samples to their neighboring classes, i.e., $Y' \leftarrow (Y+1)\%C$. In our preliminary experiment, we have actually explored three different mislabeling strategies:

1. Shifting $Y' \leftarrow (Y+1)\%C$ (adopted in Alg 1);
2. Randomly mapping $Y$ to any $Y'$ as long as $Y \neq Y'$;
3. Change the one-hot label $Y = [0, 0, \cdots, 0, 1, 0, \ldots, 0]$ to $Y = [\epsilon, \epsilon, \cdots, \epsilon, 0, \epsilon, \ldots, \epsilon]$ ($0 < \epsilon \leq 1$) in a soft-label fashion.

Surprisingly, we find the phenomenon – "*finetuning a backdoored model on a few mislabeled clean samples can cause the model to forget its regular functionality, resulting in low clean accuracy, but remarkably, its backdoor functionality remains intact, leading to a high attack success rate*" – exists regardless of the adopted mislabeling strategy choice. And as a matter of fact, BaDExpert with each of the three strategies would be similarly effective against diverse set of attacks. We finally settled at the first mislabeling choice mostly due to a stable set of hyperparameters are easier to determined than the other two strategies.

### B.4 Comparing BaDExpert with Confusion Training (Qi et al., 2023b)

Qi et al. (2023b) introduces a novel backdoor poison training set cleanser based on the technique of "confusion training", where they train an inference model jointly on the poisoned dataset and a small number of *mislabeled* clean samples (similar to our Alg (1)). Nevertheless, we highlight several critical differences between our work and theirs.

**Problems.** Qi et al. (2023b) focuses on **poisoned training set inspection** and aims at identifying poison samples within the training set. Instead, our work focuses on **identifying backdoor inputs during inference time**. The two problems have completely different setups.

**Methods.** Our method and Qi et al. (2023b)'s are different at two critical levels:

1. **Access of Information**:
   - Qi et al. (2023b) relies on the necessary *access of poisoned training samples* (i.e., requiring information about the backdoor), so that their detection model can capture the backdoor correlation.
   - Our method, on the other hand, operates independently of how the model is generated — *does not rely on any information about the backdoor samples or the poisoned dataset*, which is a significantly *more challenging* scenario.
2. **Principle**:
   - During the training on a poisoned dataset, Qi et al. (2023b) *disrupts the fitting of clean training samples using a "confusion batch" of mislabeled clean data (i.e., counteracts the gradient updates learned from the normal training samples), so that the resulting model can only capture the backdoor correlation*. They then utilize this resulting model to identify poisoned training samples by seeing which data points are correctly fitted (by comparing the model's predictions with the data points' ground-truth labels).
   - Distinguishly, our method is more related to catastrophic forgetting — we *only finetune the original backdoor model $\mathcal{M}$ on the mislabeled clean data* (without any access to poisoned samples), resulting in a backdoor expert model $\mathcal{B}$ that loses the normal functionality but retains the backdoor functionality. We then measure the agreements

Table 6: Defensive results on GTSRB (CA and ASR).

| Defenses→ | No Defense | | FP | | NC | | MOTH | | NAD | | STRIP | | AC | | Frequency | | SCALE-UP | | BaDExpert | |
|---|---|---|---|---|---|---|---|---|---|---|---|---|---|---|---|---|---|---|---|---|
| Attacks↓ | CA | ASR | CA | ASR | CA | ASR | CA | ASR | CA | ASR | CA | ASR | CA | ASR | CA | ASR | CA | ASR | CA | ASR |
| No Attack | 97.1 | - | 85.4 | - | 97.1 | - | 82.2 | - | 96.3 | - | 87.4 | - | 96.8 | - | 65.0 | - | 60.8 | - | 97.0 | - |
| **Development-Stage Attacks** | | | | | | | | | | | | | | | | | | | | |
| BadNet | 97.3 | 100.0 | 85.7 | 16.1 | 97.8 | 0.0 | 92.0 | 100.0 | 96.1 | 0.5 | 86.9 | 30.5 | 96.3 | 100.0 | 64.6 | 0.0 | 59.7 | 0.0 | 97.2 | 0.0 |
| Blend | 96.9 | 97.5 | 86.7 | 97.1 | 97.0 | 6.9 | 88.2 | 97.5 | 96.5 | 49.2 | 87.3 | 57.2 | 96.7 | 97.5 | 64.9 | 7.8 | 59.5 | 59.6 | 96.8 | 4.6 |
| Trojan | 97.1 | 100.0 | 87.1 | 98.3 | 98.1 | 0.1 | 82.0 | 100.0 | 96.4 | 34.4 | 87.4 | 64.5 | 96.9 | 100.0 | 65.1 | 0.0 | 60.4 | 95.0 | 96.9 | 3.7 |
| Dynamic | 97.1 | 100.0 | 87.0 | 100.0 | 97.4 | 34.5 | 79.0 | 100.0 | 96.9 | 64.4 | 87.4 | 37.1 | 96.8 | 100.0 | 65.0 | 12.0 | 59.3 | 24.6 | 97.0 | 0.0 |
| WaNet | 95.7 | 91.4 | 84.1 | 5.4 | 98.2 | 17.4 | 94.5 | 89.5 | 97.0 | 0.4 | 86.2 | 84.0 | 95.5 | 91.4 | 64.1 | 61.8 | 56.7 | 79.2 | 95.6 | 0.2 |
| **Post-development Attacks** | | | | | | | | | | | | | | | | | | | | |
| FT | 95.5 | 99.8 | 85.4 | 0.0 | 94.0 | 39.9 | 83.6 | 98.0 | 96.4 | 13.5 | 86.0 | 19.3 | 95.2 | 99.9 | 63.9 | 7.9 | 62.3 | 66.8 | 95.4 | 3.7 |
| TrojanNN | 96.2 | 98.4 | 84.2 | 14.8 | 96.2 | 0.2 | 78.9 | 97.7 | 96.1 | 0.3 | 86.6 | 40.5 | 96.0 | 98.4 | 64.5 | 0.0 | 62.8 | 2.1 | 96.4 | 1.9 |
| **Average** | 96.6 | 98.2 | 85.7 | 47.4 | 97.0 | 14.1 | 85.1 | 97.5 | 96.5 | 23.2 | 86.9 | 47.6 | 96.3 | 98.2 | 64.6 | 12.8 | 60.2 | 46.7 | **96.5** | **2.0** |
| CA Drop (smaller is better) | | | ↓10.9 | | ↑0.3 | | ↓11.6 | | ↓0.1 | | ↓9.7 | | ↓0.3 | | ↓32.0 | | ↓36.4 | | ↓0.1 | |
| ASR Drop (larger is better) | | | | ↓50.8 | | ↓84.0 | | ↓0.6 | | ↓74.9 | | ↓50.6 | | ↓0.0 | | ↓85.4 | | ↓51.4 | | ↓96.1 |

between the resulting backdoor expert model $\mathcal{B}$ and the original backdoor model $\mathcal{M}$, in order to identify the backdoor inputs at inference time. Notice that our approach operates without access to the ground-truth labels of inference-time inputs.

## B.5 COMPARING BADEXPERT WITH SEAM (ZHU ET AL., 2022)

Zhu et al. (2022) introduces a novel model-repairing backdoor defense (SEAM). In the first phase, they finetune the backdoored model on a small number of *mislabeled* clean samples (similar to our Alg (1)), observing that *both the CA and ASR would diminish*. In the second phase, they finetune the resultant model (after phase one) on a portion of correctly labeled samples from the training set, by which the CA will gradually recover, but the ASR will not.

Interestingly, when finetune the backdoored model on mislabeled clean samples, Zhu et al. (2022)'s observation (*both CA and ASR decrease*) seems to be different from ours (*CA drops but ASR retains*). Nevertheless, we argue that our observations are actually not contradictory to theirs.

In our method and experiments, we suggest using a conservatively small (un-)learning rate $\eta$, with which only the normal functionality degrades but the backdoor functionality retains. However, as shown in Fig 3 (and Fig 8 in Appendix C.3), when the (un-)learning rate $\eta$ is large enough (e.g., $10^{-3}$), both the normal and backdoor functionality would be lost (both CA and ASR $\rightarrow$ 0) — which corresponds to Zhu et al. (2022)'s observation. In summary, the different observations between our work and Zhu et al. (2022) are possibly due to different selections of the (un-)learning rate.

## B.6 RELATIONSHIP OF BADEXPERT WITH SHAN ET AL. (2020)

Shan et al. (2020) proposes an adversarial example detection method using "honeypots" — a trapdoor that would enforce malicious adversarial inputs to manifest a certain neural network activation-pattern signature. Their defense's key design philosophy may be subtly connected to ours, in the sense that Shan et al. (2020) detects potential adversarial examples via "similarity measurement" of model activation signatures, and we detect backdoor examples via "agreement measurement" of model prediction/confidence. However, their work's motivation, problem, and method are still largely different from ours.

## C ADDITIONAL RESULTS

### C.1 EFFECTIVENESS OF BADEXPERT ON GTSRB

Our primary results on GTSRB are shown in Table 6 and Table 7. As a general post-development defense, BaDExpert effectively defends against all attacks (average ASR = 2.0%), with a CA drop as negligible as 0.1%; Meanwhile, other baseline defenses fail against at least one backdoor attack. As a backdoor input detector, BaDExpert achieves an average 100% detection AUROC, and outperforms other baseline detectors in every scenario.

Table 7: Defensive results on GTSRB (AUROC).

| AUROC (%) | BadNet | Blend | Trojan | Dynamic | WaNet | FT | TrojanNN | Average |
|---|---|---|---|---|---|---|---|---|
| STRIP | 93.4 | 74.8 | 76.5 | 86.2 | 41.6 | 92.2 | 88.7 | 79.0 |
| AC | 52.7 | 50.2 | 32.4 | 62.7 | 34.2 | 48.3 | 30.9 | 44.5 |
| Frequency | 75.3 | 73.6 | 75.3 | 72.8 | 61.3 | 73.6 | 75.3 | 72.4 |
| SCALE-UP | 88.6 | 54.2 | 34.0 | 81.1 | 34.1 | 56.4 | 90.4 | 62.7 |
| **BaDExpert** | **100.0** | **99.9** | **100.0** | **100.0** | **100.0** | **100.0** | **99.9** | **100.0** |

Table 8: AUROC of BaDExpert with baseline (FP, NC, MOTH and NAD) repaired models as $\mathcal{M}'$ ("w/ Backdoor Expert") on CIFAR10. We use the exact ensembling decision rule in our primary experiment. Results of directly deploying the repaired models, following the soft rule described in Sec B.2, are shown in "w/o Backdoor Expert" rows. Obviously, our backdoor experts in the BaDExpert framework serve as effective augmentations (add-ons) for these baseline methods during backdoor input detection.

| Baseline as $\mathcal{M}'$ ↓ | Attacks → | BadNet | Blend | Trojan | CL | SIG | Dynamic | ISSBA | WaNet | Bpp | FT | TrojanNN | SRA | **Average** |
|---|---|---|---|---|---|---|---|---|---|---|---|---|---|---|
| FP | w/o Backdoor Expert | 10.1 | 64.3 | 64.4 | 40.5 | 90.8 | 8.3 | 97.2 | 97.3 | 99.2 | 64.1 | 66.9 | 0.2 | 58.6 |
| | w/ Backdoor Expert | 100.0 | 96.9 | 99.5 | 99.6 | 99.8 | 66.6 | 94.8 | 99.3 | 99.9 | 99.5 | 99.4 | 99.9 | **96.3** |
| NC | w/o Backdoor Expert | 99.2 | 53.4 | 99.9 | 99.9 | 66.7 | 99.5 | 98.8 | 52.7 | 99.9 | 93.2 | 99.8 | 99.9 | 88.6 |
| | w/ Backdoor Expert | 100.0 | 94.8 | 100.0 | 100.0 | 97.6 | 99.9 | 95.3 | 98.7 | 100.0 | 98.3 | 100.0 | 100.0 | **98.7** |
| MOTH | w/o Backdoor Expert | 99.7 | 32.1 | 99.0 | 99.6 | 37.0 | 94.2 | 52.6 | 95.8 | 99.4 | 63.3 | 93.2 | 99.8 | 80.5 |
| | w/ Backdoor Expert | 100.0 | 88.6 | 99.8 | 99.9 | 87.4 | 98.8 | 92.3 | 99.7 | 100.0 | 94.7 | 98.2 | 100.0 | **96.6** |
| NAD | w/o Backdoor Expert | 98.9 | 95.9 | 96.8 | 94.9 | 98.6 | 92.9 | 98.0 | 99.1 | 99.6 | 97.1 | 92.6 | 99.2 | 97.0 |
| | w/ Backdoor Expert | 100.0 | 98.6 | 99.2 | 98.6 | 99.8 | 98.0 | 95.1 | 99.8 | 100.0 | 99.5 | 97.6 | 100.0 | **98.8** |

## C.2 ENSEMBLING WITH OTHER DEFENSES

As discussed in Sec 4.2.3, we can apply any baseline-repaired models as $\mathcal{M}'$ in our BaDExpert framework, to ensemble with our backdoor experts $\mathcal{B}$. We demonstrate the ensembling results in Table 8 ("w/ Backdoor Expert" rows). For an insightful comparison, we also show the results when only the baseline-repaired model is used for backdoor input detection ("w/o Backdoor Expert" rows), following the soft decision rule for $\mathcal{M}'$ described in Sec B.2.

As shown, BaDExpert can achieve overall $\sim 99\%$ AUROCs when ensembling with NC and NAD, which align well with our major results in Table 2. When combined with FP (failed against 9 of 12 attacks in Table 1) and MOTH (failed against 6 of 12 attacks in Table 1), BaDExpert slightly degrades to $\sim 96.5\%$, due to the significant failures of the baselines themselves (which can also be told from that deploying $\mathcal{M}'$ without backdoor expert can sometimes barely achieve AUROC $< 50\%$ — worse than random guessing). Moreover, in almost all cases, BaDExpert ("w/ Backdoor Expert") achieves higher AUROCs compared to deploying $\mathcal{M}'$ alone ("w/o Backdoor Expert"). In other words, our backdoor expert models and the BaDExpert framework could serve as effective augmentations (or add-ons) to existing model-repairing backdoor defense baselines, during backdoor input detection.

## C.3 BACKDOOR EXPERTS CONSTRUCTION

For all 12 attacks evaluated in our primary experiments on CIFAR10, we visualize in Fig 8a∼8l the constructed backdoor experts' (Alg 1) CA and ASR, when we unlearn the originally backdoored model $\mathcal{M}$ with different $\eta$'s. As depicted, with a conservatively small $\eta$ (e.g. $10^{-4}$), we can always enforce the resulting backdoor expert to lose a significant amount of normal functionality (CA drop $\sim 50\%$), while retaining a similar backdoor functionality (ASR drop $\sim 0\%$). However, if we choose a large $\eta$ (e.g. 1e-3), both functionalities would be erased (both CA and ASR $\approx 0\%$). Actually, we sometimes may have to tradeoff between the maintenance of the backdoor functionality and the unlearning of the normal functionality. But overall, we can see that unlearning the backdoor functionality is *slower* than unlearning the normal functionality. More crucially, we find this phenomenon to **consistently exist across datasets and architectures** (Fig 8m∼ 8o).

An intuitive justification for such a phenomenon can be referred to and derived from Qi et al. (2023b), where the authors show that during poison training, clean samples could be forgotten faster than poison samples in the context of catastrophic forgetting (for simplified settings of training overparameterized linear model).

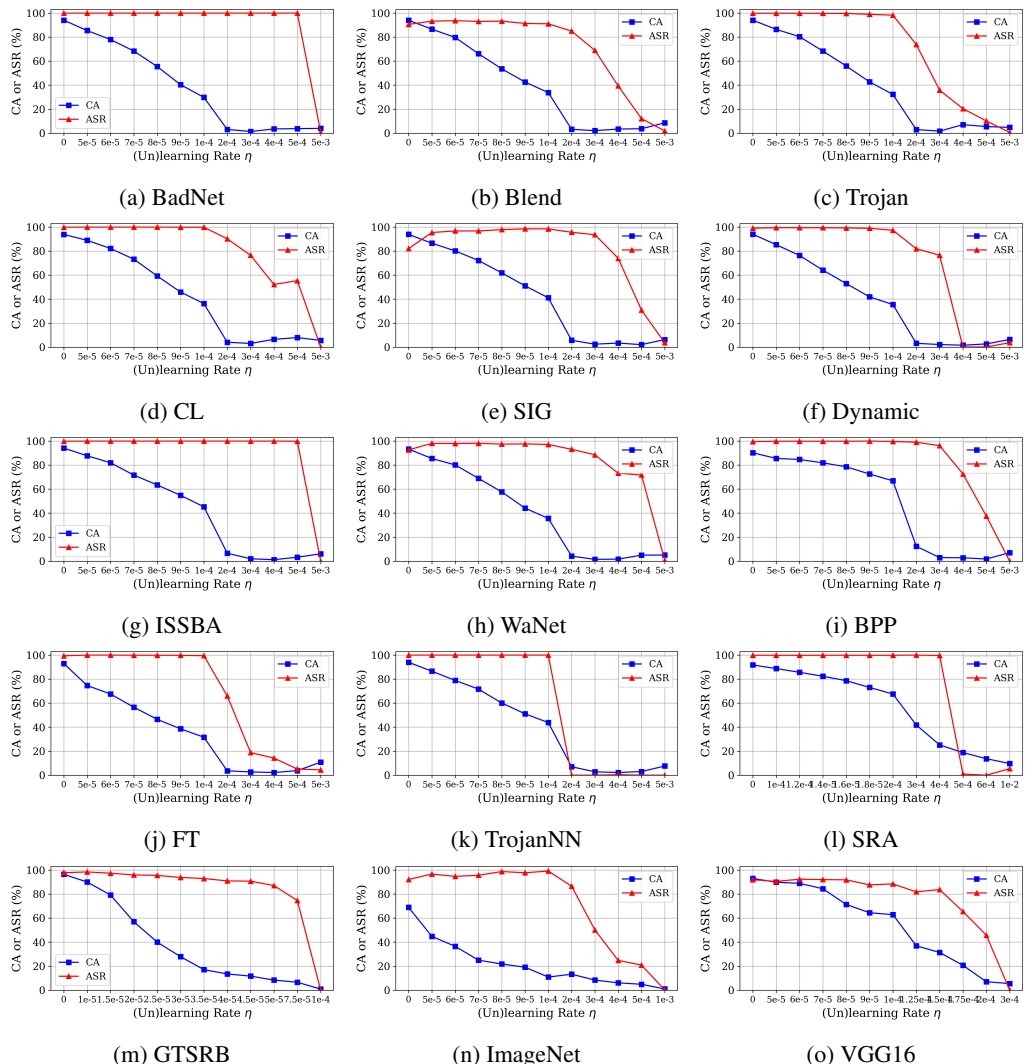

Figure 8: Unlearning curves with different $\eta$ of diverse scenarios. Fig 8a∼8l correspond to attacks conducted on CIFAR10 (ResNet18); Fig 8m and 8n correspond to Blending attacks on GTSRB and ImageNet (ResNet18), respectively; Fig 8o corresponds to the Blending attack on CIFAR10 (VGG16 instead of ResNet18).

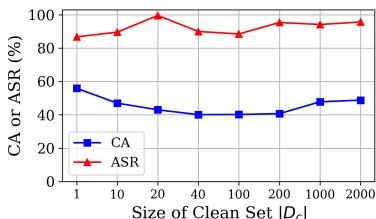

Figure 9: Backdoor experts constructed with reserved clean sets of different sizes.

Table 9: Ablation results on different (un-)learning rates $\eta$. AUROCs of BaDExpert are reported on CIFAR10.

| (Un-)learning Rate $\eta \rightarrow$ | $5*10^{-5}$ | $6*10^{-5}$ | $7*10^{-5}$ | $8*10^{-5}$ | $9*10^{-5}$ | $1*10^{-4}$ | $2*10^{-4}$ | $3*10^{-4}$ | $4*10^{-4}$ | $5*10^{-4}$ | $1*10^{-3}$ | $5*10^{-3}$ |
|---|---|---|---|---|---|---|---|---|---|---|---|---|
| BadNet | 100.0 | 100.0 | 100.0 | 100.0 | 100.0 | 100.0 | 100.0 | 100.0 | 100.0 | 100.0 | 99.9 | 90.8 |
| Blend | 97.6 | 98.1 | 98.6 | 98.7 | 99.0 | 99.2 | 99.2 | 99.0 | 99.1 | 99.2 | 95.2 | 92.5 |

Table 10: Ablation results on different poison rates. Clean accuracy (CA), attack success rate (ASR) and AUROCs of BaDExpert are reported on CIFAR10 against Blend attack.

| Poison Rate $\rightarrow$ | 0.1% | 0.2% | 0.3% | 0.4% | 0.5% | 1.0% | 5.0% | 10.0 |
|---|---|---|---|---|---|---|---|---|
| CA (before defense) | 93.6 | 94.1 | 94.1 | 93.5 | 94.2 | 93.1 | 93.9 | 94.0 |
| ASR (before defense) | 80.8 | 87.3 | 90.6 | 95.1 | 94.5 | 98.5 | 99.8 | 99.9 |
| CA (after defense) | 92.7 | 93.1 | 93.2 | 92.6 | 93.3 | 92.2 | 93.0 | 93.1 |
| ASR (after defense) | 11.8 | 3.4 | 10.8 | 7.2 | 13.3 | 8.0 | 0.3 | 0.0 |
| AUROC | 99.2 | 99.2 | 99.2 | 99.1 | 99.4 | 98.9 | 100.0 | 100.0 |

We also notice that, this property is *not sensitive w.r.t. the number of clean samples ($|D_c|$)* we have. No matter how many clean samples (from 1 or 10 to 1,000 or 2,000) we use to conduct Alg 1, with an appropriately small learning rate $\eta$ (e.g. by selecting a small $\eta$ such that the resulting CA only drops to $\sim 40\%$), we can still separate the CA and ASR by a certain extent (Fig 9).

## C.4 Ablation Studies on Different (Un-)Learning Rates $\eta$

Our method is not sensitive to the choice of (un-)learning rate $\eta$. In this ablation study, we evaluate BadExpert against BadNet and Blend attacks on CIFAR10, across a wide range of $\eta$ (from $5 \cdot 10^{-5}$ to $5 \cdot 10^{-3}$). As shown in Table 9, the AUROC of our detection remains stable across different $\eta$.

## C.5 Ablation Studies on Different Poison Rates

As shown in Table 10, BaDExpert is insensitive to the variation of poison rate. Even when the poison rate is extremely low (poison rate = $0.1\%$, equivalent to only 50 poison samples) and ASR drops to $\sim 80\%$, our BaDExpert is still manifesting near-perfect effectiveness.

## C.6 Comparing BaDExpert with STRIP and ScaleUp in Scenarios with Fewer Clean Samples

We further analyze in details how our BaDExpert would perform when there are fewer accessible clean reserved samples. We compare it alongside with two strong baseline detectors: 1) STRIP,

Table 11: Comparison of BaDExpert with STRIP and ScaleUp alongside, when they are all given equal access to different number of clean samples. AUROCs of all defenses are reported on CIFAR10 against Blend attack.

| Number of Clean Samples | Defense | BadNet | Blend |
|---|---|---|---|
| 100 | ScaleUp | 95.7 | 79.8 |
| | STRIP | 99.2 | 42.5 |
| | **BaDExpert** | **99.9** | 72.9 |
| 200 | ScaleUp | 95.7 | 79.9 |
| | STRIP | 99.3 | 44.4 |
| | **BaDExpert** | **100.0** | **91.2** |
| 400 | ScaleUp | 95.7 | 79.9 |
| | STRIP | 99.3 | 44.3 |
| | **BaDExpert** | **100.0** | **95.1** |
| 2000 | ScaleUp | 95.8 | 80.0 |
| | STRIP | 99.4 | 44.6 |
| | **BaDExpert** | **100.0** | **99.2** |

Table 12: Adaptive attack by using weakened triggers at inference time. (Blend attack on CIFAR10)

| Blending Alpha | 10% | 11% | 12% | 13% | 14% | 15% | 16% | 17% | 18% | 19% | 20% (Standard) |
|---|---|---|---|---|---|---|---|---|---|---|---|
| ASR (before defense) | 5.0 | 10.1 | 17.8 | 28.0 | 39.7 | 51.6 | 62.8 | 72.3 | 80.2 | 86.3 | 90.6 |
| ASR (after defense) | 3.0 | 5.4 | 8.5 | 11.9 | 15.3 | 18.5 | 14.8 | 16.2 | 14.0 | 15.5 | 11.4 |
| AUROC | 94.4 | 96.2 | 97.3 | 97.8 | 98.1 | 98.4 | 98.6 | 98.7 | 98.9 | 99.0 | 99.2 |

which requires only $N = 100$ clean samples by default, and 2) ScaleUp, which by default can work without clean samples (meanwhile, ScaleUp can also utilize available clean samples for its SPC value estimation). To make a fair comparison in our additional study, we assign an equal number of clean samples (100, 200, 400 and 2000) to all three defenses in each scenario.

In Table 11, we can quickly notice that **under most circumstances (number of clean samples = 200, 400 and 2,000), BaDExpert performs the best**. And when the number of clean samples is extremely restrited to 100, BaDExpert becomes less effective on Blend (72.9% AUROC), but still performs similar to ScaleUp (79.8% AUROC).

**Approaches to Obtain Reserved Clean Samples.** There exist various possiblitites to acquire such a small clean set $|D_c|$:

- A developed model usually comes with an independent set of test samples in order to validate the model' utility, the defender could simply use (a partition) of this test samples as $|D_c|$;

- Collect data (e.g. taking photos) in a secure environment;

- Manually inspect and isolate clean samples from online inputs.

Furthermore, there exists a trending line of work on isolating clean samples from a poisoned set of samples, e.g. META-SIFT (Zeng et al., 2023a). Such work can be directly applied here to aid the defender to obtain reserved clean samples. Overall, we argue that obtaining a small amount of clean samples (e.g. 200, 400, 2000) would not be a major bottleneck for defenders in practice.

## C.7 ADDITIONAL ADAPTIVE ANALYSIS DETAILS

### C.7.1 NATURAL BACKDOOR

Natural backdoors (e.g. (Zhao et al., 2022)) may exist in models trained on only clean data, which can also be considered as an adaptive backdoor attack against BaDExpert. This is because the natural backdoor functionality learned by a clean model is strongly correlated with the normal functionality – therefore, such natural backdoors also directly challenge our insight that "*backdoor functionality can be isolated from normal functionality*".

We construct such natural backdoor attacks (Zhao et al., 2022) on normally trained models (on clean data) and achieve ASR = 99.2% on CIFAR10 and ASR = 99.3% on GTSRB. We find that **BaDExpert can defend the natural backdoor attacks on both datasets with AUROC = 92.3% and AUROC = 92.8%**, respectively. While the performance shows a degradation compared to our major results in Table 2 in our paper (average AUROC = 99.0%), the >92% AUROC still reflects the nontrivial effectiveness of BaDExpert as an inference-time defense. Besides, we note that our key finding "when fintuning on mislabled clean data, the backdoor functionality would remain, while normal functionality does not" **still stands** against this attack:

- Following Alg (1) on CIFAR10, the constructed backdoor expert $\mathcal{B}$ retains a high ASR 95.3% (originally 99.2%), while the normal functionality degrades significantly (CA drops from 94.2% to 44.8%);

- On GTSRB, the observation is similar (ASR drops from 99.3% to 83.0%, CA drops from 96.8% to 41.0%).

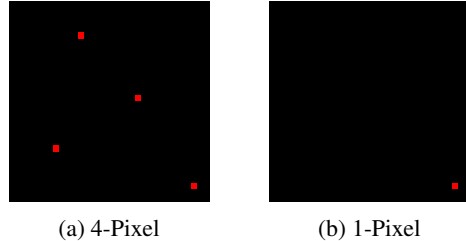

(a) 4-Pixel  (b) 1-Pixel

Figure 10: Smaller triggers demonstration.

Table 13: Adaptive attack by using smaller triggers in Fig 10. (CIFAR10)

| Attack→ | 4-Pixel | 1-Pixel |
|---|---|---|
| CA (before defense) | 94.2 | 93.7 |
| ASR (before defense) | 88.5 | 77.4 |
| CA (after defense) | 93.3 | 92.7 |
| ASR (after defense) | 14.4 | 5.4 |
| **AUROC** | 98.9 | 99.6 |

### C.7.2 ADAPTIVE ATTACK BY USING WEAKENED TRIGGERS AT INFERENCE TIME AND SMALLER TRIGGERS

Empirically, adversary may sometimes bypass existing backdoor defense methods by using a weakened version of the backdoor trigger at inference time. Therefore, we also study how our method reacts to such an adaptive attack by **decreasing the inference-time trigger blending alpha** of the Blend attack on CIFAR10. As shown in Table 12, when the adversary uses a lower blending alpha at inference time (instead of the default 20% during poisoning), the AUROC of BaDExpert indeed degrades, to as low as 94%. Nevertheless, the attack ASR drops more rapidly. Overall, we can observe a tradeoff between backdoor inputs' stealthiness (AUROC) and the attack's effectiveness (ASR). Generally speaking, **the adversary can hardly evade BaDExpert's detection by using a weaker trigger at inference time**, since the ASR will drop rapidly way before BaDExpert becomes unusable (**AUROC > 97.5% whenever ASR > 20%**).

In addition, we also study whether backdoor attackers could evade our defense via adopting smaller trigger sizes. Specifically, we evaluate BaDExpert's effectiveness against two backdoor models that are poisoned with: 1) **4-Pixel** (Fig 10a): 4 random red pixels as the backdoor trigger (0.1% poison rate); 2) **1-Pixel** (Fig 10b): 1 random red pixel as the backdoor trigger (0.5% poison rate). To ensure the backdoors are "weak", we select the minimum poison rates to make sure the backdoors are just successfully injected (non-trivial ASR). As shown in Table 13, BaDExpert can still effectively defend these attacks, suppresing the ASR to $< 14.4\%$.

### C.7.3 A TAILORED ADAPTIVE ATTACK AGAINST BaDExpert

To provide more valuable insights for future researchers / developers into our method, we also tailored a novel adaptive attack against our proposed BaDExpert.

Concretely, we assume that the adversary (who produced the backdoor model) is aware of our BaDExpert defense. Accordingly, he/she will design an evading strategy via **using an alternative trigger during inference time** (i.e., *asymmetric* backdoor trigger, which was adopted in prior work like Qi et al. (2023a)), such that the alternative trigger satisfies: 1. Can still activate the backdoor functionality of the original backdoor model $\mathcal{M}$ (i.e., achieving a high ASR) 2. Cannot activate the backdoor functionality of the original backdoor model $\mathcal{B}$ (i.e., enforcing $\mathcal{B}$ to provide a low confidence on the backdoor inputs).

Notice that **the adversary is only assumed to have access to the backdoor model** $\mathcal{M}$. The adversary, however, could also follow our BaDExpert procedure to construct a surrogate backdoor expert model $\mathcal{B}'$ (Alg 1) to the actual backdoor expert model $\mathcal{B}$ used by the victim / defender. (Empirically, we find the attack results using this surrogate backdoor expert model are similar to using the actual $\mathcal{B}$ of the

Table 14: Tailored adaptive attack against BaDExpert. (CIFAR10)

| Attack | $\lambda_2$ | 0.5 | | 0.1 | | 0.05 | | 0.01 | | |
|---|---|---|---|---|---|---|---|---|---|---|
| | $\lambda_1$ | 10 | 1 | 10 | 1 | 10 | 1 | 10 | 1 | **Average** |
| BaDExpert-Adap-BadNet | Norm | 4.8 | 1.8 | 16.6 | 5.0 | 36.7 | 5.1 | 59.8 | 5.3 | 16.9 |
| | ASR | 20.8 | 57.6 | 61.8 | 83.2 | 82.5 | 89.2 | 94.5 | 98.1 | 73.5 |
| | AUROC | 93.3 | 99.8 | 80.7 | 99.8 | 65.4 | 99.8 | 57.2 | 100.0 | 87.0 |
| BaDExpert-Adap-Blend | Norm | 11.4 | 10.2 | 32.6 | 29.5 | 44.2 | 38.6 | 59.1 | 53.5 | 34.9 |
| | ASR | 11.4 | 27.2 | 72.7 | 75.7 | 83.7 | 86.5 | 96.7 | 97.2 | 68.9 |
| | AUROC | 63.0 | 72.7 | 61.1 | 68.7 | 47.1 | 58.1 | 36.4 | 54.2 | 57.6 |

defender, if they both are trained following the same configuration. We therefore do not explicitly distinguish the notation of $\mathcal{B}$ and $\mathcal{B}'$ in the rest of this section.)

The two goals upon can be implemented via optimization on the models $\mathcal{M}$ and $\mathcal{B}$. Formally, as the adaptive adversary, we compute an alternative trigger mark $\boldsymbol{\Delta}$ and the corresponding trigger mask $\mathbf{m}$ as follow:

$$\min_{\boldsymbol{\Delta}, \mathbf{m}} \quad \text{CrossEntropyLoss}\Big(\mathcal{M}(x^*)_{\text{raw}}, t\Big) + \lambda_1 \cdot \text{Conf}_{\mathcal{B}}(t|x^*) + \lambda_2 \cdot |\mathbf{m}| \quad (x, y) \sim \mathcal{P} \quad (12)$$

$$\text{where} \quad x^* = (1 - \mathbf{m}) \odot x + \mathbf{m} \odot \boldsymbol{\Delta} \quad (13)$$

Specifically, the trigger mark $\boldsymbol{\Delta} \in \mathcal{X} = [0, 1]^{c \times w \times h}$ is a 3-dim real-value matrix (the same shape as the input $x$), while the trigger mask $\mathbf{m} \in [0, 1]^{w \times h}$ is a 2-dim real-value matrix that decides the opacity of each trigger mark pixel to mix with the corresponding input $x$s' pixel. The $\odot$ operator upon computes the Hadamard product of two matrices. (We adopt the similar trigger optimization setting in Wang et al. (2019).)

Now we explain this optimization formula:

- Minimizing the first **CrossEntropyLoss** term will help the adversary approach the first target: the alternative trigger $(\boldsymbol{\Delta}, \mathbf{m})$ can still activate the victim backdoor model $\mathcal{M}$ with a high ASR.

- The second term represents the **confidence of the backdoor expert model** $\mathcal{B}$. Minimizing it will **directly violate our core design insight**, since the resulting backdoor inputs now will not activate the backdoor functionality of the backdoor expert model $\mathcal{B}$ anymore.

- The last term, $\ell_1$ norm of the trigger mask, corresponds to the **visual stealthiness of the trigger**. The adversary usually prefers to implant a less noticeable trigger, which can be realized via minimizing the magnitude of the trigger mask $\mathbf{m}$.

The two hyperparameter $\lambda_1$ and $\lambda_2$ controls the tradeoff among these three goals.

In Table 14, we report the results of this adaptive attack ("**BaDExpert-Adap-[ORIGINAL AT-TACK]**") against our BaDExpert defense when $\lambda_1$ and $\lambda_2$ are selected differently. To fully study the potential tradeoff between the adaptive attack effectiveness and its defense evasiveness, we report alongside 1) the adaptive trigger **Norm** $|\mathbf{m}|$; 2) the attack success rate (**ASR**); 3) **AUROC** of BaDExpert against the tailored adaptive attack.

As shown in Table 14, we can observe that the tailored adaptive attack could effectively diminish the effectiveness of BaDexpert – AUROC becomes as low as $57.2\%$ against BaDExpert-Adap-BadNet and $36.4\%$ against BaDExpert-Adap-Blend. Meanwhile, we can also notice two trends from the table:

1. As $\lambda_2$ becomes larger (with $\lambda_1$ fixed), the **ASR** becomes higher, while the defense **AUROC** decreases. However, the increasing evasiveness of the adaptive attack comes with a price: the magnitude of the trigger mask (**Norm**) also increases – i.e., the backdoor attack becomes less stealthy.

2. As $\lambda_1$ becomes higher (with $\lambda_2$ fixed), the defense **AUROC** effectively degrades. However: 1) the **ASR** drops, and 2) the backdoor attack becomes stealthier since **Norm** becomes larger.

Table 15: Comparing BaDExpert with additional baseline defenses: ANP Wu & Wang (2021), I-BAU Zeng et al. (2021a), ABL Li et al. (2021a), AWM Chai & Chen (2022), RNP Li et al. (2023) and CD Huang et al. (2023). CA and ASR are reported on CIFAR10.

| Attack | | No Defense | ANP | I-BAU | ABL | AWM | RNP | CD | **BaDExpert (Ours)** |
|--------|-----|-----------|------|-------|------|------|------|------|-----------------------|
| BadNet | CA | 94.1 | 80.8 | 88.0 | 92.5 | 92.7 | 70.0 | 78.6 | **93.1** |
| | ASR | 100.0 | 0.4 | 0.8 | 11.7 | 4.8 | 18.9 | 2.9 | **0.0** |
| Blend | CA | 94.1 | 83.2 | 90.4 | 91.9 | 88.9 | 77.9 | 79.0 | **93.1** |
| | ASR | 90.6 | 16.8 | 16.6 | 11.8 | 29.7 | 19.6 | 72.8 | **11.4** |

In brief, we can see that **the adaptive attack can indeed restrict BaDExpert's performance, at the cost of either attack effectiveness (lower ASR) or stealthiness (higher Norm)**. Overall, BaDExpert still performs nontrivially in most scenarios – 87.0% average AUROC against BaDExpert-Adap-BadNet and 57.6% against BaDExpert-Adap-Blend.

## C.8 COMPARING BADEXPERT WITH ADDITIONAL BASELINES

In Table 15, we further compare BaDExpert with 6 additional recent backdoor defense baselines: ANP (Wu & Wang, 2021), I-BAU (Zeng et al., 2021a), ABL Li et al. (2021a), AWM Chai & Chen (2022), RNP Li et al. (2023) and CD Huang et al. (2023), w.r.t. attack success rate and clean accuracy. Among them: 1) ABL is a poison suppresion method that happens in the model development stage; 2) ANP, I-BAU, RNP and AWM are post-development model-reparing defenses; 3) CD is originally a poisoned training dataset cleanser, and we adapted it to a backdoor input detector. As shown in Table 15, BaDExpert outperforms all of them, achieving lower ASR and higher CA.

## C.9 STANDARD DEVIATIONS OF MAJOR EXPERIMENTS

Standard deviations of our major experiments Table 1, 2, 6 and 7 are shown in Table 16, 17, 18 and 19.

Table 16: Standard deviation of Table 1.

| Defenses→ | No Defense | | FP | | NC | | MOTH | | NAD | | STRIP | | AC | | Frequency | | SCALE-UP | | **BaDExpert** | |
|-----------|------|------|------|------|------|------|------|------|------|------|------|------|------|------|------|------|------|------|------|------|
| **Attacks ↓** | CA | ASR | CA | ASR | CA | ASR | CA | ASR | CA | ASR | CA | ASR | CA | ASR | CA | ASR | CA | ASR | CA | ASR |
| No Attack | 0.3 | - | 1.1 | - | 1.4 | - | 0.3 | - | 0.6 | - | 0.3 | - | 0.4 | - | 0.2 | - | 0.0 | - | 0.3 | - |
| **Development-Stage Attacks** | | | | | | | | | | | | | | | | | | | | |
| BadNet | 0.1 | 0.0 | 0.4 | 0.0 | 0.2 | 2.4 | 0.4 | 0.2 | 0.5 | 0.4 | 0.1 | 0.1 | 0.1 | 0.1 | 0.1 | 0.0 | 0.1 | 0.0 | 0.1 | 0.0 |
| Blend | 0.1 | 1.3 | 0.7 | 1.4 | 0.6 | 1.7 | 0.1 | 11.5 | 0.4 | 4.7 | 0.1 | 1.1 | 0.6 | 14.4 | 0.1 | 0.2 | 0.3 | 2.8 | 0.1 | 6.5 |
| Trojan | 0.2 | 0.1 | 0.7 | 36.0 | 0.2 | 0.5 | 0.5 | 2.1 | 0.2 | 2.8 | 0.2 | 15.3 | 0.2 | 0.4 | 0.2 | 0.0 | 2.5 | 5.8 | 0.2 | 3.3 |
| CL | 0.2 | 0.0 | 0.9 | 7.3 | 0.2 | 0.5 | 0.2 | 0.5 | 0.4 | 3.5 | 0.2 | 3.7 | 0.2 | 0.3 | 0.3 | 0.0 | 1.1 | 0.0 | 0.2 | 7.1 |
| SIG | 0.4 | 0.7 | 2.6 | 25.8 | 0.4 | 7.2 | 0.2 | 50.9 | 0.2 | 6.0 | 0.4 | 3.2 | 0.4 | 12.1 | 0.4 | 0.2 | 0.5 | 24.6 | 0.4 | 1.7 |
| Dynamic | 0.2 | 0.2 | 1.3 | 11.8 | 0.1 | 5.0 | 0.5 | 41.2 | 0.5 | 1.3 | 0.3 | 4.5 | 4.3 | 43.2 | 0.2 | 0.0 | 0.9 | 0.2 | 0.2 | 5.4 |
| ISSBA | 0.3 | 0.0 | 0.2 | 0.2 | 0.3 | 2.0 | 0.3 | 34.3 | 0.6 | 0.3 | 0.2 | 2.1 | 0.3 | 0.1 | 0.3 | 0.0 | 0.1 | 0.1 | 0.3 | 0.5 |
| WaNet | 0.6 | 0.9 | 1.8 | 2.6 | 0.8 | 20.3 | 0.4 | 10.1 | 1.3 | 0.4 | 0.5 | 0.7 | 0.6 | 3.8 | 0.5 | 0.9 | 0.1 | 1.6 | 0.5 | 0.3 |
| BPP | 0.5 | 0.2 | 0.4 | 8.8 | 0.3 | 57.3 | 0.2 | 1.5 | 0.7 | 0.2 | 0.4 | 3.7 | 0.5 | 0.6 | 0.6 | 0.0 | 4.7 | 10.2 | 0.5 | 0.1 |
| **Post-development Attacks** | | | | | | | | | | | | | | | | | | | | |
| FT | 0.2 | 0.2 | 0.8 | 4.6 | 0.4 | 16.2 | 0.5 | 5.1 | 0.8 | 1.0 | 0.3 | 3.3 | 0.2 | 9.6 | 0.2 | 0.2 | 0.2 | 7.4 | 0.2 | 1.8 |
| TrojanNN | 0.2 | 0.0 | 0.4 | 9.6 | 0.1 | 0.3 | 0.4 | 23.4 | 0.3 | 7.7 | 0.2 | 0.1 | 0.2 | 0.0 | 0.2 | 0.0 | 1.4 | 0.0 | 0.3 | 6.6 |
| SRA | 1.4 | 0.0 | 1.2 | 0.0 | 0.4 | 0.6 | 0.4 | 0.5 | 2.6 | 0.4 | 1.2 | 0.9 | 1.4 | 0.0 | 1.0 | 0.0 | 0.8 | 5.3 | 1.4 | 0.4 |

Table 17: Standard deviation of Table 2

| **AUROC** (%) | BadNet | Blend | Trojan | CL | SIG | Dynamic | ISSBA | WaNet | Bpp | FT | TrojanNN | SRA |
|---------------|--------|-------|--------|------|------|---------|-------|-------|------|------|----------|------|
| STRIP | 0.2 | 2.2 | 10.6 | 3.4 | 6.8 | 2.0 | 1.4 | 0.8 | 13.1 | 1.4 | 0.3 | 0.5 |
| AC | 0.0 | 10.1 | 0.1 | 0.0 | 2.0 | 7.2 | 2.8 | 0.4 | 0.2 | 15.3 | 0.0 | 0.0 |
| Frequency | 0.1 | 0.1 | 0.0 | 0.1 | 0.0 | 0.1 | 0.1 | 0.2 | 0.1 | 0.1 | 0.1 | 0.1 |
| SCALE-UP | 0.6 | 1.5 | 4.4 | 0.7 | 1.8 | 0.4 | 0.4 | 0.8 | 8.9 | 2.6 | 0.8 | 2.4 |
| **BaDExpert** | 0.0 | 0.1 | 0.2 | 0.4 | 0.1 | 0.1 | 1.0 | 0.0 | 0.0 | 0.1 | 0.6 | 0.0 |

Table 18: Standard Deviation of Table 6.

| Defenses→ | No Defense | | FP | | NC | | MOTH | | NAD | | STRIP | | AC | | Frequency | | SCALE-UP | | **BaDExpert** | |
|---|---|---|---|---|---|---|---|---|---|---|---|---|---|---|---|---|---|---|---|---|
| Attacks↓ | CA | ASR | CA | ASR | CA | ASR | CA | ASR | CA | ASR | CA | ASR | CA | ASR | CA | ASR | CA | ASR | CA | ASR |
| No Attack | 0.4 | - | 1.8 | - | 0.5 | - | 3.7 | - | 0.5 | - | 0.4 | - | 0.4 | - | 0.3 | - | 3.5 | - | 0.4 | - |
| **Development-Stage Attacks** | | | | | | | | | | | | | | | | | | | | |
| BadNet | 0.6 | 0.0 | 1.7 | 11.5 | 1.0 | 0.0 | 0.6 | 0.0 | 0.3 | 0.1 | 0.8 | 1.7 | 0.8 | 0.1 | 0.6 | 0.0 | 2.7 | 0.1 | 0.6 | 0.1 |
| Blend | 0.5 | 0.3 | 0.7 | 0.8 | 0.1 | 1.1 | 1.2 | 1.1 | 0.3 | 13.0 | 0.5 | 5.6 | 0.5 | 0.2 | 0.4 | 0.1 | 0.9 | 4.3 | 0.5 | 2.9 |
| Trojan | 0.2 | 0.0 | 0.2 | 1.1 | 0.3 | 0.1 | 3.5 | 0.0 | 0.6 | 56.8 | 0.2 | 3.9 | 0.2 | 0.0 | 0.2 | 0.0 | 1.1 | 2.2 | 0.0 | 2.2 |
| Dynamic | 0.1 | 0.0 | 0.1 | 0.0 | 0.6 | 56.7 | 3.5 | 0.0 | 0.3 | 49.2 | 0.1 | 2.9 | 0.1 | 0.0 | 0.1 | 0.0 | 1.6 | 6.6 | 0.1 | 0.1 |
| WaNet | 0.2 | 0.5 | 2.1 | 5.9 | 0.2 | 20.3 | 0.5 | 0.4 | 0.1 | 0.1 | 0.1 | 0.6 | 0.2 | 0.5 | 0.2 | 0.4 | 0.8 | 0.3 | 0.2 | 0.0 |
| **Post-development Attacks** | | | | | | | | | | | | | | | | | | | | |
| FT | 0.6 | 0.1 | 0.5 | 0.0 | 0.6 | 19.9 | 4.7 | 1.3 | 0.2 | 18.2 | 0.6 | 4.0 | 0.6 | 0.1 | 0.4 | 0.0 | 1.1 | 1.7 | 0.6 | 4.7 |
| TrojanNN | 0.7 | 2.4 | 1.0 | 14.7 | 0.7 | 0.3 | 7.0 | 3.2 | 0.3 | 0.2 | 0.6 | 10.5 | 0.7 | 2.4 | 0.5 | 0.0 | 1.0 | 3.3 | 0.8 | 1.8 |

Table 19: Standard deviation of Table 7.

| **AUROC** (%) | BadNet | Blend | Trojan | Dynamic | WaNet | FT | TrojanNN |
|---|---|---|---|---|---|---|---|
| STRIP | 1.8 | 2.6 | 2.7 | 1.6 | 1.1 | 1.9 | 3.7 |
| AC | 29.9 | 7.6 | 0.6 | 9.8 | 2.2 | 6.2 | 0.2 |
| Frequency | 0.1 | 0.0 | 0.0 | 0.1 | 0.1 | 0.0 | 0.0 |
| SCALE-UP | 0.6 | 1.9 | 4.9 | 2.8 | 0.3 | 1.2 | 1.5 |
| **BaDExpert** | 0.0 | 0.2 | 0.0 | 0.0 | 0.0 | 0.0 | 0.0 |

