# OpenReview forum: "BaDExpert: Extracting Backdoor Functionality for Accurate Backdoor Input Detection"
_ICLR.cc/2024/Conference — ICLR 2024 poster_

### Official Review · Reviewer_79zr · 2023-10-19

**Soundness:** 3 good
**Presentation:** 3 good
**Contribution:** 2 fair
**Rating:** 6
**Confidence:** 4

**Summary:**

This paper proposed a backdoor defence method called BaDExpert. The main motivation is that normal data can be quickly forgotten with finetuning on inconsistent labels, while backdoor data does not. Based on this characteristic, the proposed method can extract a backdoor functionality of the model, which can only correctly classify backdoor data. Experiments show that the proposed method can effectively detect and defend against backdoor attacks.

**Strengths:**

- The proposed method is simple, straightforward and effective. Extracting the backdoor functionality is a new idea in backdoor attack research.
- The empirical evaluations are very comprehensive, including different attacks, datasets, model architectures, and adaptive attacks. Results show it is effective against existing attacks and also demonstrated its limits under adaptive cases.

**Weaknesses:**

- The proposed method relies on several procedures, each with different hyperparameters. However, the authors provide an ablation study in each component, with no overall insights/guides for applying such a method in real-world applications. In real applications, the defender does not know for sure that backdoor attacks exist in their data, so it might not be easy to find these suitable hyperparameters.
- The comprehensive experimental results are appreciated. It could be more comprehensive to compare with recent defence methods such as ABL [1], and detection methods [2,3].
- The method mainly focuses on the detection of backdoor samples (end of section 3). It could help to further clarify what happens after. Results in Table 1 focus on the CA/ASR. It is not clear what happens to the detected samples in order to obtain these results.

[1] Li, Yige, et al. "Anti-backdoor learning: Training clean models on poisoned data." Advances in Neural Information Processing Systems (2021).\
[2] Pan, Minzhou, et al. "ASSET: Robust Backdoor Data Detection Across a Multiplicity of Deep Learning Paradigms." USENIX Security Symposium (2023).\
[3] Huang, Hanxun, et al. "Distilling Cognitive Backdoor Patterns within an Image." The Eleventh International Conference on Learning Representations (2023).\

**Questions:**

No further questions, please address the weakness section.

---

> ### Author Response · Authors · 2023-11-18
> **Response to Reviewer 79zr (Part 1)**
>
> We thank the reviewer for acknowledging our proposed method and experimental evaluations. We have incorporated the reviewer's suggestions, and hope our following clarifications could further resolve the reviewer's remaining concerns:
>
> ---
>
> ### **Q1: Hyperparameter selection.**
>
> We totally agree with the reviewer that it's important to provide a general insight/guidance to real-world practitioners, in order to help them select an appropriate set of hyperparameters. In our submitted draft, we have briefly discussed how to select key hyperparameters in Appendix A.1 and A.2. To futher clarify, we make the following arguments to show that probing for an appropriate set of hyperparameters for BaDExpert is not difficult in practice.
> * **Fine-tuning rate $\eta’$.** In Appendix A.1.1, we have provided a brief ***rule of thumb*** to guide practitioners to select the fine-tuning learning rate $\eta'$, by observing the clean accuracy (CA) trend of the model during fine-tuning. Specifically, we observe that clean finetuning can best diminish the backdoor functionality with a large learning rate, where the finetuned model’s CA drops to ∼ $0$% in the first one or two epochs, and then recovers to a significant value in the following epochs.
> * **Unlearning rate $\eta$**. As studied in Appendix C.4, we have shown that our method is consistently effective across a wide range of $\eta$ (from $5\cdot 10^{-5}$ to $5\cdot 10^{-3}$). Nevertheless, we understand the reviewer's concern, that practitioners in the real world might need certain guidance to identify this coarse-grained range of $\eta$.
>
>     Similar to $\eta'$, a simple ***rule of thumb for $\eta$*** could be: starting from an $\eta$ that is orders of magnitude smaller than the $\eta'$ used for clean finetuning, keep tuning $\eta$ until observing the model's clean accuracy (CA) diminishes by a certain extent (but does not completely vanish, i.e., worse than random guess). This rule of thumb helps practitioners idenfity an approximate range of $\eta$ according to the model's CA drop magnitude, which can be derived from Fig 8 -- we can see the backdoor functionality is usually preserved when $\eta$ is conservatively small, in the sense that CA has not degraded to the random-guess level.

---

> > ### Author Response · Authors · 2023-11-18
> > **Response to Reviewer 79zr (Part 2)**
> >
> > ### **Q2: Additional comparisons with recent baselines.**
> >
> > As suggested by the reviewer, we have supplemented the results comparing BaDExpert with ABL[1] and CD[4] (together with additional baselines suggested by other reviewers) in Rebuttal-Table 1. Particulary, we note that:
> > * ABL[1] is a poison-suppression defense that requires control of the training process (pre-development), which is actually out of the scope considered by our work (post-development). We simply report the CA/ASR results of the anti-backdoor learned models.
> > * CD[4] is a technique originally for poisoned training set inspection (i.e., detecting poison samples within the training dataset). In order to compare CD with BaDExpert, we adapt it to an inference-time backdoor input detector.
> > * ASSET[5] is also a poisoned training set inspection method. We did not include it because it cannot be applied/adapted to our inference-time backdoor input detection setting.
> >
> > As shown in Rebuttal-Table 1 below, BaDExpert outperforms both ABL and CD, achieving lower ASR while retaining higher CA. These results are also updated in our paper revision (Appendix C.8).
> >
> > **Rebuttal-Table 1: Comparing BaDExpert with additional baselines.** (CA for Clean Accuracy, ASR for Attack Success Rate) For fair comparison: we follow the suggested hyperparameters and only tune them to approach the officially reported results; when reporting RNP results, we stop pruning once ASR drops to below $20$%; when reporting CD results, we follow their suggested and default threshold selection strategy. As shown, **BaDExpert outperforms all these baselines w.r.t. both CA and ASR**.
> >
> >
> > | Attack | | No Defense | ABL[1] | AWM[2] | RNP[3] | CD[4] | BaDExpert |
> > | -------- | -------- | -------- | -------- | -------- | -------- | -------- | -------- |
> > | **BadNet** | CA  | 94.1 | 92.5 | 92.7 | 70.0 | 78.6 | **93.1** |
> > |             | ASR | 100.0 | 11.7 | 4.8 | 18.9 | 2.9 | **0.0** |
> > | **Blend**  | CA  | 94.1 | 91.9 | 88.9 | 77.9 | 79.0 | **93.1** |
> > |             | ASR | 90.6 | 11.8 | 29.7 | 19.6 | 72.8 | **11.4** |
> >
> > ---
> >
> > ### **Q3: Clarifying what happens after the detection of backdoor samples.**
> >
> > We apologize for any unclarity in our experimental setting w.r.t. the reported CA/ASR results of BaDExpert. Indeed, our work only focuses on **detecting** backdoor inputs (and does not study how to obtain correct classification retuls for them). In Sec 4.1 "Metrics" paragraph, we have stated how we report the CA & ASR for BaDExpert (and other detector-based defenses), in order to fairly compare with other non-detector-based defenses: "*we interpret correctly filtered backdoor inputs as successfully thwarted attacks, while falsely filtered clean inputs are considered erroneous predictions*". Formally and specifically:
> > 1. $\text{ASR}=\mathbb P_{(x,y)\sim \mathcal{P}} [\mathcal M(\mathcal T(x)) = t \wedge \mathtt{BID}(\mathcal T(x)) = 0]$: the ASR is reported by considering all correctly detected (rejected) backdoor inputs as failed attack attempts, and thus not inlcuded in the ASR numerator;
> > 2. $\text{CA}=\mathbb P_{(x,y)\sim \mathcal{P}} [\mathcal M(x) = y \wedge \mathtt{BID}(x) = 0]$: the CA is reported by considering all falsely detected (rejected) benign inputs as misclassification, and thus not inlcuded in the CA numerator.
> >
> >
> > ---
> > [1] Li, Yige, Xixiang Lyu, Nodens Koren, Lingjuan Lyu, Bo Li, and Xingjun Ma. "Anti-backdoor learning: Training clean models on poisoned data." Advances in Neural Information Processing Systems 34 (2021): 14900-14912.
> >
> > [2] Chai, Shuwen, and Jinghui Chen. "One-shot neural backdoor erasing via adversarial weight masking." Advances in Neural Information Processing Systems 35 (2022): 22285-22299.
> >
> > [3] Li, Yige, Xixiang Lyu, Xingjun Ma, Nodens Koren, Lingjuan Lyu, Bo Li, and Yu-Gang Jiang. "Reconstructive Neuron Pruning for Backdoor Defense." arXiv preprint arXiv:2305.14876 (2023).
> >
> > [4] Huang, Hanxun, Xingjun Ma, Sarah Erfani, and James Bailey. "Distilling Cognitive Backdoor Patterns within an Image." arXiv preprint arXiv:2301.10908 (2023).
> >
> > [5] Pan, Minzhou, Yi Zeng, Lingjuan Lyu, Xue Lin, and Ruoxi Jia. "ASSET: Robust Backdoor Data Detection Across a Multiplicity of Deep Learning Paradigms." arXiv preprint arXiv:2302.11408 (2023).

---

> > ### Comment · Reviewer_79zr · 2023-11-22
> >
> > Thanks for the clarifications. They addressed concerns Q2/Q3. The reviewer would suggest the authors add these results to the paper and provide additional details/analysis regarding these new experiments.
> >
> > Regarding Q1, the reviewer is not convinced by the clarification. In practice, the defender should not know what is the current CA. This is also mentioned by reviewer 59E7.

---

> > > ### Author Response · Authors · 2023-11-22
> > > **Followup Response to Reviewer 79zr**
> > >
> > > Dear Reviewer 79zr,
> > >
> > > We appreciate your feedback to our rebuttal response very much, and are glad to hear about the additional concern. Regarding Q2/Q3, we have already updated these results and analysis into our revised paper, and will keep working on related details.
> > >
> > > Regarding the clean accuracy (CA) knowledge in practice, we would like to provide more clarification as follow:
> > > - First, the defender can estimate the model's CA on his/her reserved clean samples. And that **a local clean test set uniquely preserved to evaluate the model's utility (CA) is a necessary and practical assumption**:
> > >     - This is a **necessary** assumption, since the model owner is going to deploy the model for his/her own downstream task, and he/she must make sure the model has reached such capability, i.e., high CA.
> > >     - We also believe this is a **practical** assumption, given that nowadays there exist standard test sets available for diverse CV tasks. Furthermore, such a clean test set can also be obtained by withholding a small fraction of the clean samples $D_c$ (we originally preserved for unlearning / finetuning).
> > > - Second, **the assumption of "knowing what is the current CA" has been made in most prior works, either explicitly or implicitly**. As some examples:
> > >     - In NAD [1], the distillation and erasing learning rates also need to be selected according to the model's CA.
> > >     - In ABL [2], the isolation learning rate and the unlearning rate need to be probed via observing the model's CA.
> > >     - In NC [3], a clean test set must be known in order to estimate the reversed trigger's effectiveness. During the trigger unlearning stage, the model's CA must be available to help practitioners select a proper unlearning rate.
> > >
> > > [1] "Neural attention distillation: Erasing backdoor triggers from deep neural networks." International Conference on Learning Representations (2021).
> > >
> > > [2] "Anti-backdoor learning: Training clean models on poisoned data." Advances in Neural Information Processing Systems (2021).
> > >
> > > [3] "Neural cleanse: Identifying and mitigating backdoor attacks in neural networks." IEEE Symposium on Security and Privacy (2019).
> > >
> > > Thanks,
> > >
> > > Authors

---

> > > > ### Comment · Reviewer_79zr · 2023-11-23
> > > >
> > > > Thanks for the clarification. Reserving a portion of the data for evaluating CA is reasonable and practical. The reviewer suggests the authors add experiments to verify this setting in the next revision, e.g. reserve a portion of the training set on CIFAR.
> > > >
> > > > Although the reviewer still has some concerns about the practicality of the proposed method, considering the interesting idea of extracting the backdoor functionality and the rebuttal discussion, the reviewer believes the merit outweighs the weaknesses and will increase the rating to 6.

---

> > > > > ### Author Response · Authors · 2023-11-23
> > > > >
> > > > > Dear Reviewer 79zr,
> > > > >
> > > > > We would like to thank you for the acknowledgement of our work and all your engagement during the review & rebuttal stages!
> > > > >
> > > > > Thanks,
> > > > >
> > > > > Authors

---

> ### Author Response · Authors · 2023-11-21
> **A Gentle Reminder of the Final Feedback**
>
> Dear Reviewer 79zr,
>
> We would like to thank the reviewer for the helpful suggestions. We hope our previous response has adequately resolved your concerns regarding *applying our method in the real world* and comparing with *additional baselines*. As the deadline of ICLR rebuttal period is approaching, we look forward to hearing your feedback on our response, and would be happy to clarify any additional questions.
>
> Best,
>
> Authors

---

### Official Review · Reviewer_xRnn · 2023-10-26

**Soundness:** 4 excellent
**Presentation:** 4 excellent
**Contribution:** 3 good
**Rating:** 8
**Confidence:** 5

**Summary:**

A novel backdoor defense BaDExpert is proposed in this paper. BaDExpert is designed to distinguish test instances with the backdoor trigger from benign test instances. The key idea is to fine-tune a "backdoor-expert" model with only the backdoor functionality so that benign test instances will be poorly recognized. Thus, test instances that are differently classified by the backdoored model and the backdoor-expert model are deemed benign; otherwise, a test instance is deemed to contain the trigger.

**Strengths:**

1) The method is well-motivated and the idea is novel.

2) The experiments are thorough, involving many attack settings and covering many SOTA baselines.

3) The presentation is excellent.

**Weaknesses:**

More comparisons with existing works regarding the methodology can be included.

**Questions:**

1. Is the design philosophy of BaDExpert related to [1]? In [1], adversarial examples are detected by encouraging the model to carry malicious behaviors such as a backdoor.

[1] Shan et al, Gotta Catch 'Em All: Using Honeypots to Catch Adversarial Attacks on Neural Networks, 2019.

2. Can BaDExpert outperform [2] which also detects malicious inputs?

[2] Li et al, Test-time detection of backdoor triggers for poisoned deep neural networks, 2022.

---

> ### Author Response · Authors · 2023-11-18
> **Response to Reviewer xRnn**
>
> We thank the reviewer for the high appreciation of our work. We hope our following supplementary results and clarifications could further resolve the reviewer's remaining concerns.
>
> ---
>
> ### **Q1: Additional comparisons with recent baselines.**
>
> During the rebuttal periord, we have supplemented results comparing BaDExpert with several additional baselines [1-4]. As shown in Rebuttal-Table 1, BaDExpert outperforms all these baseline defenses, achieving higher CA and lower ASR. These results are also updated in our paper revision (Appendix C.8).
>
> Regarding the reviewer's suggested baseline [5], we have included it as a related work in our revision (see updated Sec 5). But due to time and resource limitations, we were yet not able to incorporate the experimental results of [5]; we will try to include the results in the future revisions.
>
> **Rebuttal-Table 1: Comparing BaDExpert with additional baselines.** (CA for Clean Accuracy, ASR for Attack Success Rate) For fair comparison: we follow the suggested hyperparameters and only tune them to approach the officially reported results; when reporting RNP results, we stop pruning once ASR drops to below $20$%; when reporting CD results, we follow their suggested and default threshold selection strategy. As shown, **BaDExpert outperforms all these baselines w.r.t. both CA and ASR**.
>
>
> | Attack | | No Defense | ABL[1] | AWM[2] | RNP[3] | CD[4] | BaDExpert |
> | -------- | -------- | -------- | -------- | -------- | -------- | -------- | -------- |
> | **BadNet** | CA  | 94.1 | 92.5 | 92.7 | 70.0 | 78.6 | **93.1** |
> |             | ASR | 100.0 | 11.7 | 4.8 | 18.9 | 2.9 | **0.0** |
> | **Blend**  | CA  | 94.1 | 91.9 | 88.9 | 77.9 | 79.0 | **93.1** |
> |             | ASR | 90.6 | 11.8 | 29.7 | 19.6 | 72.8 | **11.4** |
>
> ---
>
> ### **Q2: Relationship to Shan et al.[6].**
>
> We thank the reviewer for the pointer to [6], where the authors propose an adversarial example detection method using "honeypots" --- a trapdoor that would enforce malicious adversarial inputs to manifest a certain neural network activation-pattern signature. We also found that their key design philosophy to be subtly connected to ours, in the sense that [6] detect potential adversarial examples via "activation signature similarity measurement", and we detect backdoor examples via "model prediction/confidence agreement measurement". However, their motivation, problem, and method are still largely different from ours, thus we did not discuss this work in our previous submitted version. To make sure we cover related work comprehensively, we have added a separate Appendix section (B.6) to discuss the connection between our design to [6] in our revision.
>
> ---
>
> [1] Li, Yige, Xixiang Lyu, Nodens Koren, Lingjuan Lyu, Bo Li, and Xingjun Ma. "Anti-backdoor learning: Training clean models on poisoned data." Advances in Neural Information Processing Systems 34 (2021): 14900-14912.
>
> [2] Chai, Shuwen, and Jinghui Chen. "One-shot neural backdoor erasing via adversarial weight masking." Advances in Neural Information Processing Systems 35 (2022): 22285-22299.
>
> [3] Li, Yige, Xixiang Lyu, Xingjun Ma, Nodens Koren, Lingjuan Lyu, Bo Li, and Yu-Gang Jiang. "Reconstructive Neuron Pruning for Backdoor Defense." arXiv preprint arXiv:2305.14876 (2023).
>
> [4] Huang, Hanxun, Xingjun Ma, Sarah Erfani, and James Bailey. "Distilling Cognitive Backdoor Patterns within an Image." arXiv preprint arXiv:2301.10908 (2023).
>
> [5] Li, Xi, Zhen Xiang, David J. Miller, and George Kesidis. "Test-time detection of backdoor triggers for poisoned deep neural networks." In ICASSP 2022-2022 IEEE International Conference on Acoustics, Speech and Signal Processing (ICASSP), pp. 3333-3337. IEEE, 2022.
>
> [6] Shan, Shawn, Emily Wenger, Bolun Wang, Bo Li, Haitao Zheng, and Ben Y. Zhao. "Gotta catch'em all: Using honeypots to catch adversarial attacks on neural networks." In Proceedings of the 2020 ACM SIGSAC Conference on Computer and Communications Security, pp. 67-83. 2020.

---

### Official Review · Reviewer_59E7 · 2023-10-30

**Soundness:** 3 good
**Presentation:** 3 good
**Contribution:** 3 good
**Rating:** 5
**Confidence:** 4

**Summary:**

This paper proposes a novel backoor detection approach: It firstly finetune the backdoor model over a small subset of mislabeled samples to remove the benign functionality while preserving the backdoor-related functionality. Then, badexpert detects the backdoor examples based on the agreement between the backdoor expert and backdoored model. The effectiveness of the propsed method is tested on both small  (CIFAR-10 and GTSRB) and large dataset (ImageNet), CNN and ViT.

**Strengths:**

1 This paper is easy to follow.

2 This paper is well written.

3 Appreciate the solid experiments shown in the experimental section.  The authors demonstrate the novel performance of Badexpert in multiple datasets and architectures.

**Weaknesses:**

1 Admitting the effectiveness of the proposed method, I think the practicality of Badexpert is limited. As shown in Appendix A, the optimal learning rate could vary across datasets: ($\eta=10^{-4}$ for CIFAR-10 and $\eta=2.5\cdot10^{-5}$ for GTSRB). Even for the same dataset,  the optimal $\eta$ could be different across different architectures:   $\eta=10^{-4}$ for ResNet18 and  $\eta=10^{-6}$ for pretrained vit_b_16. Considering the tremendous hyperparameter required for Badexpert, I think the overall guidlines for how to choose hyperparameter are needed to help Badexpert better defend against  potential risks.

2 BadExpert depends on the stong mapping from the trigger to the pre-defined behaviour. I firmly believe, as long as the backdoor attack is weak enough (decreasing the poison rate/ the size of trigger/ the blend rate), will potentially leads to the Badexpert unsuccessful.  Sincerely, I hope to further discuss with the authors when encountering the above situations. Table 10 shows parts of the results, but not enough from my view. In addition, I think ASR and CA is a more appropriate metric to indicate the performance of BaDExpert instead of AUROC: Combing Table 1 and Table 2, BaDExpert obtains 11.4% ASR against blend attack. However, the AUROC is 99.2% which is quite close to 100%.

3 The chosen of hyperparemter $\eta'$ could also be unpractical in real world. In reality, only the small subset of clean images is available for defenders. Therefore, they have little knowledge about which $\eta'$will meets the requirement of Badexpert: the CA of the finetuned model's CA first drops to ~ $0\%$ and recovers to a significant value in the following epochs. The defender dooesn't exactly know what the CA of current model is. Therefore, the requirement of Badexpert may be too ideal.

4 Some of the baselines are missing. For example, AWM [1] and ABL [2].

**Questions:**

1 Table 1 shows that BadExpert is relatively weak to defend against Blend or Dynamic attack. Can you explain the reason behind this phenomenon?

For other questions, please refer to the weakness section.

[1] One-shot Neural Backdoor Erasing via Adversarial Weight Masking

[2] Anti-Backdoor Learning: Training Clean Models on Poisoned Data

**Details Of Ethics Concerns:**

Is the **ethics statement section** considered as the part of main text for the submitted paper? The ethics statement appears in the page 10. However, according to ICLR's rule, the main text of the submission is limited to 9 pages with unlimited additional pages for citations or appendixes.

---

> ### Author Response · Authors · 2023-11-18
> **Response to Reviewer 59E7 (Part 1)**
>
> We thank the reviewer for acknowledging our proposed method, experiments, and paper writing. We have incorporated the reviewer's suggestions, and hope our following clarifications could further resolve the reviewer's remaining concerns:
>
> ---
>
> ### **Q1: Hyperparameter selection.**
>
> We totally understand the reviewer's concern regarding our method's practicality of hyperparameter choosing. We hereby clarify our practical guidances for hyperparameters selection in respond to the reviewer's concern, showing that probing for an appropriate set of hyperparameters for BaDExpert is not difficult in practice.
> * **Fine-tuning rate $\eta’$.** In Appendix A.1.1, we provide a brief rule of thumb to select the fine-tuning learning rate $\eta'$ by observing the clean accuracy (CA) trend of the model. Specifically, we observe that clean finetuning can best diminish the backdoor functionality with a large learning rate, where the finetuned model’s CA drops to ∼$0$% in the first one or two epochs, and then recovers to a significant value in the following epochs.
>
>     Indeed, we implicitly assume the model deployer (defender) has certain knowledge of the model $\mathcal M$'s clean utility (or CA), since he/she is going to deploy the model for his/her own downstream task (and must make sure the model has such capability, i.e., high CA). We believe this is not a "too ideal" assumption, given that nowadays a standard test set is often available for diverse CV tasks, and that the clean utility of the model to be deployed is rather crucial to the model deployer.
> * **Unlearning rate $\eta$**. To clarify, we would not claim our selection of $\eta$ is "optimal" in each setting -- we only adjust the $\eta$ in a coarse-grained way such that the unlearning process (Alg 1) does not collapse when the model architecture is different. In fact, as studied in Appendix C.4, we have shown that our method is consistently and similarly effective across a wide range of $\eta$ (from $5\cdot 10^{-5}$ to $5\cdot 10^{-3}$). Nevertheless, we understand the reviewer's concern, that practitioners in the real world might need certain guidance to identify this coarse-grained range of $\eta$.
>
>     Similar to $\eta'$, a simple **rule of thumb for $\eta$** could be: starting from an $\eta$ that is orders of magnitude smaller than the $\eta'$ used for clean finetuning, keep tuning $\eta$ until observing the model's clean accuracy (CA) diminishes by a certain extent (but does not completely vanish, i.e., worse than random guess). This rule of thumb helps practitioners idenfity an approximate range of $\eta$ according to the model's CA drop magnitude, which can be derived from Fig 8 -- we can see the backdoor functionality is usually preserved when $\eta$ is conservatively small, in the sense that CA has not degraded to the random-guess level.

---

> > ### Author Response · Authors · 2023-11-18
> > **Response to Reviewer 59E7 (Part 2)**
> >
> > ### **Q2: BaDExpert against weaker backdoor attacks** (lower poison rate / smaller blending rate / smaller trigger size).
> >
> > We thank the reviewer for questioning about whether weaker backdoor attacks would hinder our defense. As noticed by the reviewer, we have briefly discussed when the attacker tries to make their attack stealthier via lowering the poisoning rate (Table 10, or Rebuttal Table 1 below) and the blending rate (Table 12, or Rebuttal Table 2 below).
> >
> > In order to showcase our defense's effectiveness against weaker backdoor attacks, we further supplement the following three tables (lower poison rate / smaller blending rate / smaller trigger size, respectively). The defense results are reported in the reviewer's suggested metric, CA and ASR (where FPR is also fixed to $1$%, in consistent with our setting across the entire paper).
> >
> >
> > * In Rebuttal-Table 1, we can see BaDExpert is consistently effective across a wide range of posion rate, from $0.1$% (50 poison samples) to $10.0$% (5000 poison samples).
> > * In Rebuttal-Table 2, BaDExpert also effectively suppresses the ASR no matter what blending rate (from $10$% to $20$%) the attacker employs.
> > * In Rebuttal-Table 3, we study two extreme scenarios where the attacker adopts small trigger sizes (together with low poison rates). As shown, BaDExpert still successfully defends these attack attempts.
> >
> > These ablation studies are also updated in our paper revision (Table 10, 12, 13 and Fig 10).
> >
> > **Rebuttal-Table 1: Ablation results of different poison rates. (Blend attack on CIFAR10, FPR=$1$%)** As shown, BaDExpert is insensitive to the variation of poison rate. Even when the poison rate is extremely low (poison rate = 0.1%, equivalent to only 50 poison samples) and ASR drops to ~80%, our BaDExpert still successfully suppresses the ASR to $11.8$%.
> >
> > | Poison Rate $\rightarrow$ | 0.1% | 0.2% | 0.3% | 0.4% | 0.5% | 1.0% | 5.0% | 10.0% |
> > | -------- | -------- | -------- | -------- | -------- | -------- | -------- | -------- | -------- |
> > | CA  (before defense) | 93.6 | 94.1 | 94.1 | 93.5 | 94.2 | 93.1 | 93.9 | 94.0 |
> > | ASR (before defense) | 80.8 | 87.3 | 90.6 | 95.1 | 94.5 | 98.5 | 99.8 | 99.9 |
> > | **CA (after defense)** | 92.7 | 93.1 | 93.2 | 92.6 | 93.3 | 92.2 | 93.0 | 93.1 |
> > | **ASR (after defense)** | 11.8 | 3.4 | 10.8 | 7.2 | 13.3 | 8.0 | 0.3 | 0.0 |
> >
> > **Rebuttal-Table 2: Ablation results of using weakened triggers at inference time. (Blend attack on CIFAR10, FPR=$1$%)** As shown, when the adversary try to bypass our defense by using a lower blending rate at inference time (instead of the default 20% during poisoning), BaDExpert can still consistently suppress the ASR to below $20$%.
> >
> > | Blending Rate $\rightarrow$ | 10% | 11% | 12% | 13% | 14% | 15% | 16% | 17% | 18% | 19% | 20% (Standard) |
> > | -------- | -------- | -------- | -------- | -------- | -------- | -------- | -------- | -------- | -------- | -------- | -------- |
> > | ASR (before defense) | 5.0 | 10.1 | 17.8 | 28.0 | 39.7 | 51.6 | 62.8 | 72.3 | 80.2 | 86.3 | 90.6 |
> > | **ASR (after defense)** | 3.0 | 5.4 | 8.5 | 11.9 | 15.3 | 18.5 | 14.8 | 16.2 | 14.0 | 15.5 | 11.4 |
> >
> > **Rebuttal-Table 3: Ablation results of backdoor attacks with small trigger sizes. (CIFAR10, FPR=$1$%)** We further study BaDExpert's effectiveness against two backdoor models that are poisoned with small triggers: 1) 4-Pixel: 4 random red pixels as the backdoor trigger ($0.1$% poison rate); 2) 1-Pixel: 1 random red pixel as the backdoor trigger ($0.5$% poison rate). Demonstrations of these triggers are available in our revised paper (Fig 10). To ensure the backdoors are "weak", we select the minimum poison rates to make sure the backdoors are just successfully injected (non-trivial ASR). As shown, BaDExpert can still effectively defend these attacks, suppresing the ASR to $<14.4$%.
> >
> > | Attack | 4-Pixel | 1-Pixel |
> > | -------- | -------- | -------- |
> > | CA  (before defense) | 94.2 | 93.7 |
> > | ASR (before defense) | 88.5 | 77.4 |
> > | **CA (after defense)** | 93.3 | 92.7 |
> > | **ASR (after defense)** | **14.4** | **5.4** |

---

> > > ### Author Response · Authors · 2023-11-18
> > > **Response to Reviewer 59E7 (Part 3)**
> > >
> > > ### **Q3: Additional comparisons with recent baselines.**
> > >
> > > We note that ABL is a poison-suppression defense that requires control of the training process (pre-development), which is actually out of the scope considered by our work (post-development). Neverthelss, as suggested by the reviewer, we have supplemented the results comparing BaDExpert with ABL[1] and AWM[2] (together with additional baselines suggested by other reviewers) in Rebuttal-Table 1. As shown, BaDExpert outperforms both ABL and AWM, achieving higher CA and lower ASR. These results are also updated in our paper revision (Appendix C.8).
> > >
> > > **Rebuttal-Table 4: Comparing BaDExpert with additional baselines.** (CA for Clean Accuracy, ASR for Attack Success Rate) For fair comparison: we follow the suggested hyperparameters and only tune them to approach the officially reported results; when reporting RNP results, we stop pruning once ASR drops to below $20$%; when reporting CD results, we follow their suggested and default threshold selection strategy. As shown, **BaDExpert outperforms all these baselines w.r.t. both CA and ASR**.
> > >
> > >
> > > | Attack | | No Defense | ABL[1] | AWM[2] | RNP[3] | CD[4] | BaDExpert |
> > > | -------- | -------- | -------- | -------- | -------- | -------- | -------- | -------- |
> > > | **BadNet** | CA  | 94.1 | 92.5 | 92.7 | 70.0 | 78.6 | **93.1** |
> > > |             | ASR | 100.0 | 11.7 | 4.8 | 18.9 | 2.9 | **0.0** |
> > > | **Blend**  | CA  | 94.1 | 91.9 | 88.9 | 77.9 | 79.0 | **93.1** |
> > > |             | ASR | 90.6 | 11.8 | 29.7 | 19.6 | 72.8 | **11.4** |
> > >
> > > ---
> > >
> > > ### **Q4: BaDExpert's degraded performance against Blend and Dynamic.**
> > >
> > > According to Table 2, we can see the AUROC against Blend and Dynamic are actually at the average level ($99.2$% and $99.1$%, respectively) -- implying that BaDExpert is averagely good at *separating* clean and backdooor inputs against these two attacks (compared to other attacks). However, in Table 1, the defended ASR for them are indeed slightly higher than the defended ASR of other attacks. We suspect the reasons behind could be one of the following:
> > > 1. The underlying backdoor correlation is less robust / more brittle (when we fine-tune the models on mislabeled clean samples), compared to other attacks. And therefore, our backdoor extraction cannot completely isolate the backdoor functionality (i.e., the backdoro expert model $\mathcal B$ becomes the straggler).
> > > 2. The underlying backdoor correlation is more robust / too persistant (when we fine-tune the models on correctly labeled clean samples), compared to other attacks. And therefore, the standard fine-tuning process cannot satisfactorily diminish the backdoor functionality (i.e., the auxiliary model $\mathcal M'$ becomes the straggler).
> > >
> > > But overall, we could still say that BaDExpert is effective, and doesn't demonstrate significant degradation against these two attacks. Once we allow a slightly higher FPR (e.g., $2$%), the defended ASR would be further suppressed to $8.4$% and $7.1$%. Moreover, according to our experiments on GTSRB (Table 6, where the ASR are $4.6$% and $0.0$%, respectively), BaDExpert also does not perform noticeably worse on these two particular attacks.
> > >
> > > ---
> > >
> > > ### **Q5: Concern regarding ICLR page limitation.**
> > >
> > > According to [ICLR author guide](https://iclr.cc/Conferences/2024/AuthorGuide): "The optional ethic statement will not count toward the page limit, but should not be more than 1 page."
> > >
> > > ---
> > > [1] Li, Yige, Xixiang Lyu, Nodens Koren, Lingjuan Lyu, Bo Li, and Xingjun Ma. "Anti-backdoor learning: Training clean models on poisoned data." Advances in Neural Information Processing Systems 34 (2021): 14900-14912.
> > >
> > > [2] Chai, Shuwen, and Jinghui Chen. "One-shot neural backdoor erasing via adversarial weight masking." Advances in Neural Information Processing Systems 35 (2022): 22285-22299.
> > >
> > > [3] Li, Yige, Xixiang Lyu, Xingjun Ma, Nodens Koren, Lingjuan Lyu, Bo Li, and Yu-Gang Jiang. "Reconstructive Neuron Pruning for Backdoor Defense." arXiv preprint arXiv:2305.14876 (2023).
> > >
> > > [4] Huang, Hanxun, Xingjun Ma, Sarah Erfani, and James Bailey. "Distilling Cognitive Backdoor Patterns within an Image." arXiv preprint arXiv:2301.10908 (2023).

---

> ### Author Response · Authors · 2023-11-21
> **A Gentle Reminder of the Final Feedback**
>
> Dear Reviewer 59E7,
>
> We would like to thank the reviewer for the helpful suggestions. We hope our previous response has adequately resolved your concerns regarding our method's *practicality*, our defense against *weak backdoor attacks*, and comparing with *additional baselines*. As the deadline of ICLR rebuttal period is approaching, we look forward to hearing your feedback on our response, and would be happy to clarify any additional questions.
>
> Best,
>
> Authors

---

> > ### Comment · Reviewer_59E7 · 2023-11-22
> > **Thank you for your rebuttal**
> >
> > Dear authors,
> >
> > Thank you for your retailed rebuttal. I still have concerns about the hyperparameter settings of BadExpert. In addition, I notice that the hyperparameter of the baseline defense, e.g. NAD, are consistent cross different attacks. I worry about the fairness of comparison in Table 1.
> >
> > Best,
> >
> > Reviewer 59E7

---

> > > ### Author Response · Authors · 2023-11-22
> > > **Followup Response to Reviewer 59E7**
> > >
> > > Dear Reviewer 59E7,
> > >
> > > We appreciate your kind feedback to our rebuttal response very much!
> > >
> > > Regarding your additional concern about fairness w.r.t. hyperparameters of the baseline defenses, we would like to provide a more detailed clarification as follow:
> > > * On the one hand, **when running NAD on different datasets, its hyperparameters are also different**. Following their official implementation and ensuring that NAD performs stably on different datasets, NAD's learning rate is set to $0.1$ on CIFAR10 (Table 1) and $0.02$ on GTSRB (Table 6). We apologize that we have not provided such details in our paper, and will clarify this in our next revision.
> > > * On the other hand, **both NAD and our defense (BaDExpert) use a fixed set of hyperparameters across different attacks on the same dataset**. In Table 1, i.e., our major experiments on CIFAR10, we fix our hyperparameters $\eta$ to $10^{-4}$ and $\eta'$ to $0.1$ across all attacks.
> > >
> > > In summary, we believe that the comparison we demonstrate in Table 1 (and other Tables) is fair, in the sense that we use a consistent hyperparameters across different attacks, just as other baselines do.
> > >
> > > Best,
> > >
> > > Authors

---

> ### Author Response · Authors · 2023-11-23
> **Looking forward to hear from you regarding your additional concern**
>
> Dear Reviewer 59E7,
>
> As it's the last few hours to the end of ICLR rebuttal period, we look forward to hearing from you regarding whether our followup response resolves your additional concern. Again, we are grateful for your engagement during the review and rebuttal periods!
>
> Best,
>
> Authors

---

### Official Review · Reviewer_Xe68 · 2023-11-01

**Soundness:** 2 fair
**Presentation:** 3 good
**Contribution:** 2 fair
**Rating:** 6
**Confidence:** 2

**Summary:**

This paper introduces BaDExpert, an innovative defense mechanism against backdoor attacks targeting deep neural networks (DNNs). The defense is built upon the concept of extracting the backdoor functionality from a backdoored model to create a backdoor expert model. The backdoor expert model is then used to detect backdoor inputs during model inference. BaDExpert's efficacy is showcased across various datasets and model architectures, highlighting its impressive performance in terms of AUROC, a significant reduction in Attack Success Rate (ASR), and a minimal decline in clean accuracy (CA).

**Strengths:**

- The paper introduces a novel approach for defending against backdoor attacks by extracting the backdoor functionality from a backdoored model.
- The paper provides a well-structured explanation of the methodology.
- The paper presents extensive experimental results on multiple datasets and model architectures, demonstrating the effectiveness of BaDExpert.

**Weaknesses:**

- The paper lacks in-depth theoretical analysis to support the proposed method.

- The technique may not perform optimally when applied to models that haven't been backdoored.

- The experimental section seems to omit comparisons with certain recent relevant works.

**Questions:**

(1) A core tenet of the proposed method is that fine-tuning on a small set of mislabeled clean data can isolate the backdoor functionality. While this paper attempts to validate the idea through experimentation, providing a rigorous theoretical analysis would bolster the method's credibility.

(2)  In real-world scenarios, after acquiring a model online, it's often uncertain whether it has been backdoored. If the model is a benign one, there would be a disagreement between the outputs of model \mathcal{M} and \mathcal{B} (on the left side of Figure 2). This divergence could potentially hinder BaDExpert's performance.

(3) Could this paper elucidate the time complexity of the proposed method and compare it with methods like I-BAU? Given that this technique necessitates model fine-tuning, there are concerns about its efficiency.

(4) It seems that some recent published related works are missing to be compared in the paper. For example,  [1] presents defense results that are on par with those in this paper, reporting an ASR of 5.03 and a CA of 92.18 for the CIFAR10 dataset.

(5) Publicly releasing the code would facilitate better reproducibility and peer verification, enhancing the paper's value.

If the authors could solve some concerns mentioned above, the reviewer would reconsider the rating.

[1] Li, Y., Lyu, X., Ma, X., Koren, N., Lyu, L., Li, B., and Jiang, Y.G., 2023. *Reconstructive Neuron Pruning for Backdoor Defense*. arXiv preprint arXiv:2305.14876.

---

> ### Author Response · Authors · 2023-11-18
> **Response to Reviewer Xe68 (Part 1)**
>
> We thank the reviewer's acknowledgement of our work, together with the constructive suggestions and questions. We have incorporated the reviewer's suggestions, and hope our following clarifications may help resolve the raised concerns & questions:
>
> ---
>
> ### **Q1: Lack of theoretical analysis.**
>
> We sincerely appreciate the reviewer's suggestion on theoretically analyzing our observation that "fine-tuning on a small set of mislabeled clean data can isolate the backdoor functionality". In our work, we have provided theoretical insight into the *decision rule* (Neyman-Pearson Lemma in Remark 1, Sec 4.2). Mathematically analyzing the effect of *learning (fine-tuning) dynamics* of neural networks, on the other hand, is rather technically difficult --- researchers are often simplifying DNNs into overparameterized linear models to do this. These simplified analysis can only provide more empirical intuitions behind the methods, but not rigorous guarantees.
>
> In our work, in order to provide sturdy support for our method's "credibility", we tried our best to conduct various experiments, as many as possible. Regarding the observation of "*fine-tuning on a small set of mislabeled clean data can isolate the backdoor functionality*", we have shown its pervasiveness across a big arrary of attacks (Fig 8). Regarding the overall effectiveness of our method, BaDExpert, we have validated its effectiveness across different attacks, datasets, architectures, and accompanied with various ablation studies. Moreover, we have studied its "worst-case" effectiveness against a tailored adaptive attack. All these empirical efforts jointly confirm our method's credibility.
>
> ---
>
> ### **Q2: BaDExpert's performance on clean (benign) models**.
>
> In Table 1 "No Attack" row, we have shown the case when the model $\mathcal M$ is benign (CA = $93.1$%). In case of potential misunderstanding, we clarify the occasion when $\mathcal M$ is not backdoored more specifically:
>
> 1. *Remember that "backdoor input" is only meaningful when the model is embedded with a corresponding backdoor.* So when the model $\mathcal M$ is benign, there would be no concept as "backdoor input" or "backdoor functionality"; every input can simply be considered as a "clean input". As a result, Fig 2 (left) has no meaning in this context -- there is no attacker, and the "patched inputs" shown in Fig 2 (left) can just be considered as benign inputs with watermarks.
> 3. The extracted "backdoor expert model" $\mathcal B$ is just a DNN that "forgets its benign task" -- making incorrect predictions and disagreeing with $\mathcal M$ (in most cases). Therefore, by looking at Fig 2 (right), we can come to the conclusion: BaDExpert would accept most benign images (i.e., low false positive rates).
>
> In summary, we argue that BaDExpert should be similarly effective, no matter the model $\mathcal M$ is backdoored or benign.
>
> P.S.: In Sec 4.4 and Appendix C.8.1, we have also considered the **Natural** backdoor attack (which is not a traditional backdoor attack setting, more similar to the universal adversarial attack setting; our arguments above do not include this) against *benign models*. We found that BaDExpert is still effective at detecting these Natural backdoor inputs (AUROC $>92$%), and our key finding that "*when finetuning on mislabeled clean data, the backdoor functionality would remain, while normal functionality does not*" still holds in general.

---

> > ### Author Response · Authors · 2023-11-18
> > **Response to Reviewer Xe68 (Part 2)**
> >
> > ### **Q3: Additional comparisons with recent baselines.**
> >
> > As suggested by the reviewer, we have supplemented results comparing BaDExpert with RNP[3] (together with additional baselines suggested by other reviewers). As shown in Rebuttal-Table 1, BaDExpert outperforms RNP, achieving higher CA and lower ASR. These results are also updated in our paper revision (Appendix C.8).
> >
> > $^\star$**Regarding the reviewer's quote "RNP reports an average ASR of 5.03 and CA of 92.18 on CIFAR10".** We note that RNP reports these results with a **default poisoning rate of $5$%**, which is far larger than our default poisoning rate $0.3$%. Such discrepant settings may lead to the different reported results of RNP. In fact, the RNP authors also note in the Sec 5 that "*RNP faces a noticeable challenge when defending against backdoor attacks with low poisoning rates (e.g., the poisoning rate ≤ 1%).*" Moreover, their results in Table 12 (RNP against lower-poisoning-rate attackers) match our reported RNP results; and we have also validated that RNP can achieve the similar reported performance at the $5$% poisoning rate.
> >
> > **Rebuttal-Table 1: Comparing BaDExpert with additional baselines.** (CA for Clean Accuracy, ASR for Attack Success Rate) For fair comparison: we follow the suggested hyperparameters and only tune them to approach the officially reported results; when reporting RNP results, we stop pruning once ASR drops to below $20$%; when reporting CD results, we follow their suggested and default threshold selection strategy. As shown, **BaDExpert outperforms all these baselines w.r.t. both CA and ASR**.
> >
> >
> > | Attack | | No Defense | ABL[1] | AWM[2] | RNP[3] | CD[4] | BaDExpert |
> > | -------- | -------- | -------- | -------- | -------- | -------- | -------- | -------- |
> > | **BadNet** | CA  | 94.1 | 92.5 | 92.7 | 70.0 | 78.6 | **93.1** |
> > |             | ASR | 100.0 | 11.7 | 4.8 | 18.9 | 2.9 | **0.0** |
> > | **Blend**  | CA  | 94.1 | 91.9 | 88.9 | 77.9 | 79.0 | **93.1** |
> > |             | ASR | 90.6 | 11.8 | 29.7 | 19.6 | 72.8 | **11.4** |
> >
> > ---
> >
> > ### **Q4: Method efficiency concern.**
> >
> > * The time complexity analysis in I-BAU[5] points out its time complexity to be $\tilde O(K \cdot \vartheta \cdot \tilde O(\theta))$, where $K$ is the epochs (rounds) of their unlearning algorithm, $\vartheta$ is the number of the iterations of their fixed-point solver, $O(\theta)$ is the time cost for training a DNN via backprop on the reserved clean set for one epoch. In practice, they choose $K=3$ and $\vartheta=5$, i.e. approximately $15\cdot \tilde O(\theta)$, costing $26$ GPU seconds in our environment (ResNet18 on CIFAR10).
> > * Similarly, we could establish the time complexity of BaDExpert as $\tilde O((K_1 + K_2) \cdot \tilde O(\theta))$, where $K_1$ is the number of unlearning epochs (to obtain $\mathcal B$), and $K_2$ is the fine-tuning epochs (to obtain $\mathcal M'$). In practice, we choose $K_1=1$ and $K_2=10$, i.e. approximately $11\cdot \tilde O(\theta)$, costing $23$ GPU seconds in our environment (ResNet18 on CIFAR10).
> >
> > As shown, our method's efficiency outperforms I-BAU both in principle and in practice.
> >
> > ---
> >
> > ### **Q5: Code releasing.**
> >
> > Thanks for the suggestion regarding publicly releasing our code! We will definitely share our code on GitHub after the peer-review process for reproducibility and verification purpose.
> >
> > ---
> >
> > [1] Li, Yige, Xixiang Lyu, Nodens Koren, Lingjuan Lyu, Bo Li, and Xingjun Ma. "Anti-backdoor learning: Training clean models on poisoned data." Advances in Neural Information Processing Systems 34 (2021): 14900-14912.
> >
> > [2] Chai, Shuwen, and Jinghui Chen. "One-shot neural backdoor erasing via adversarial weight masking." Advances in Neural Information Processing Systems 35 (2022): 22285-22299.
> >
> > [3] Li, Yige, Xixiang Lyu, Xingjun Ma, Nodens Koren, Lingjuan Lyu, Bo Li, and Yu-Gang Jiang. "Reconstructive Neuron Pruning for Backdoor Defense." arXiv preprint arXiv:2305.14876 (2023).
> >
> > [4] Huang, Hanxun, Xingjun Ma, Sarah Erfani, and James Bailey. "Distilling Cognitive Backdoor Patterns within an Image." arXiv preprint arXiv:2301.10908 (2023).
> >
> > [5] Zeng, Yi, Si Chen, Won Park, Z. Morley Mao, Ming Jin, and Ruoxi Jia. "Adversarial unlearning of backdoors via implicit hypergradient." arXiv preprint arXiv:2110.03735 (2021).

---

> ### Author Response · Authors · 2023-11-21
> **A Gentle Reminder of the Final Feedback**
>
> Dear Reviewer Xe68,
>
> We would like to thank the reviewer for the helpful suggestions. We hope our previous response has adequately resolved your concerns regarding our method's performance on *clean / benign models*, comparing with *additional baselines*, and *efficiency (time complexity)*. As the deadline of ICLR rebuttal period is approaching, we look forward to hearing your feedback on our response, and would be happy to clarify any additional questions.
>
> Best,
>
> Authors

---

> ### Comment · Reviewer_Xe68 · 2023-11-22
> **Thanks for the response**
>
> Dear Authors,
>
> Thanks a lot for the response. It solved some of my concerns such as the code and new baselines. However, I still have some concerns regarding the experiment settings and the model settings.
>
> First, for new baselines, it would be better if the paper can include more details, such as the experiment settings (i.e., which dataset, hyper-parameters), and evaluation on more datasets. If the results are obtained from CIFAR dataset, the ASR of the original model seems to be low for me (I am unsure, since it can vary for different settings, that is why I feel I need some clarifications on the settings).
>
> Second, based on my understanding, I still feel the method might not be robust enough. it seems the paper requires some knowledge about the CA/model estimation before the defenses, which is not available in most of the cases (similar as pointed out by reviewer 79zr). That is also why I wonder if the proposed method can work perfectly on different clean models. Even though estimation could be possible, it is hard to really implement it. For example, how to sample the reserved test set, what size, and how often for the testing are difficult to decide, and the numbers vary for different attacks as well as datasets.

---

> > ### Author Response · Authors · 2023-11-22
> > **Followup Response to Reviewer Xe68**
> >
> > Dear Reviewer Xe68,
> >
> > It is great to know that our previous rebuttal response helps solve some of your concerns! Regarding your additional concerns, please let us clarify as follow.
> >
> > ---
> >
> > ### Additional Concern #1: More details of the additional comparison
> >
> > We apologize that we did not include **detailed configurations / settings of the comparison** of our method with baseline defenses. These additional experiments focus on attacks on CIFAR10 (ResNet18). When implementing these additional baselines, we follow the suggested hyperparameters in either the corresponding papers or the official repositories; we also ensure they are working as effective as possible in our setting by tuning these hyperparameters according to the eventual defense effectiveness.
> > * Specifically, when comparing with RNP, the unlearning stage contains $20$ epochs (lr = $0.01$) with an early stopping threshold of CA=$20$%, the recovering stage contains $20$ epochs (lr = $0.2$).
> >
> > Regarding the **discrepancy in ASR**: in Table 1, we report the average results (ASR & CA) over three repetitive experiments, corresponding to 3 independently curated backdoor models. And when reporting the supplementary results during rebuttal, we experiment on one backdoor model of the three, and thus the initial (No Defense) CA/ASR of Rebuttal-Table 1 may be slightly different (from the Table 1). Nevertheless, **to ensure that we do not bias toward our method during comparing, we run these baselines multiple times and report their best defense results possible.**
> >
> > We will include these configuration / setting details in our paper's next revision. We also thank the reviewer's suggestion on comparing our method with additional baselines on additional datasets. In our current paper, we already provide extensive comparisons on both CIFAR10 (Table 1) and GTSRB (Table 6) with multiple baselines. Nevertheless, we will try to extend this additional comparison also on GTSRB, in our future paper revision.
> >
> > ---
> >
> > ### Additional Concern #2: CA/model estimation
> >
> > We argue that **a local clean test set uniquely preserved to evaluate the model's utility (CA) is a necessary and practical assumption**.
> > - This is a **necessary** assumption, since the model owner is going to deploy the model for his/her own downstream task, and he/she must make sure the model has reached such capability, i.e., high CA.
> > - We also believe this is a **practical** assumption, given that nowadays there exist standard test sets available for diverse CV tasks. Furthermore, such a clean test set can also be obtained by withholding a small fraction of the clean samples $D_c$ (we originally preserved for unlearning / finetuning).
> >
> > Also, **the assumption of CA/model estimation is commonly made by most previous works**:
> > * **Noticeably, in the reviewer's mentioned baseline RNP [1]**, the ability to estimate CA is also necessary to help select their unlearning and recovering LR. Our method does not make any more assumption / require any more knowledge than theirs.
> > * In NAD [2], the distillation and erasing learning rates also need to be selected according to the model's CA.
> > * In ABL [3], the isolation learning rate and the unlearning rate need to be probed via observing the model's CA.
> > * In NC [4], a clean test set must be known in order to estimate the reversed trigger's effectiveness. During the trigger unlearning stage, the model's CA must be available to help practitioners select a proper unlearning rate.
> >
> > To sum up, we believe that knowing the CA / estimating the model utility is a natural and practical assumption to make, which has been adopted in most prior works. And our work requires only the same level of knowledge / assumptions regarding this.
> >
> > ---
> >
> > [1] Li, Y., Lyu, X., Ma, X., Koren, N., Lyu, L., Li, B., and Jiang, Y.G., 2023. Reconstructive Neuron Pruning for Backdoor Defense. arXiv preprint arXiv:2305.14876.
> >
> > [2] "Neural attention distillation: Erasing backdoor triggers from deep neural networks." International Conference on Learning Representations (2021).
> >
> > [3] "Anti-backdoor learning: Training clean models on poisoned data." Advances in Neural Information Processing Systems (2021).
> >
> > [4] "Neural cleanse: Identifying and mitigating backdoor attacks in neural networks." IEEE Symposium on Security and Privacy (2019).
> >
> > Thanks,
> >
> > Authors

---

> ### Comment · Reviewer_Xe68 · 2023-11-23
> **Response to the rebuttal**
>
> Dear authors,
>
> Thanks for the quick response. The rebuttal addressed most of my questions, but I still have some minor concerns about the details of the experiment, including the setup for each experiment, as well as the specific parameter choices, sensitivities, and assumptions about the defenders. Hence, I will keep my score for now.

---

> > ### Author Response · Authors · 2023-11-23
> >
> > Dear Reviewer Xe68,
> >
> > We appreciate your feedback very much. Following your suggestions, in our paper's future revision, we will elaborate more on:
> > 1. **Experimental setup** (dataset, model architecture, etc.) in every table;
> > 2. **Hyperparameters** of all baseline defenses and our method in every different setting (which has been briefly described in Appendix A);
> > 3. Highlighting our method's **insensitivity** to hyperparameter choices (which has been briefly discussed in Appendix C.4);
> > 4. **Assumptions** of the defender we are considering (including the natural and practical assumption to evaluate model CA on a test set).
> >
> > Again, we sincerely thank the reviewer for acknowledging our work. We are more than grateful for your active engagement during the rebuttal period, together with your valuable contribution brought to us to improve our work.
> >
> > Thanks,
> >
> > Authors

---

### Author Response · Authors · 2023-11-20
**Thanks to all reviewers. Don't hesitate to let us know if there are any additional questions/concerns!**

We would like to express our sincere gratitude towards all the reviewers' efforts to review our work and to provide valuable feedback. Particularly, we are deeply encouraged by the reviewers' acknowledgments of our *proposed method's novelty*, *extensive experimental evaluations*, and our *paper writing*.

Should you have any additional questions or require further clarification of our proposed backdoor defense (BaDExpert), please let us know anytime! We are more than happy to address them before the rebuttal period ends.

---

### Meta-Review · Area_Chair_FzdV · 2023-12-12

**Metareview:**

Most reviewers are positive about this paper. Hence, the paper will be accepted. The main weaknesses are: assumptions about the defenders, details of the experiments,  the hyperparameter settings and the fairness of comparison in Table 1, and practicality of the proposed method.

**Justification For Why Not Higher Score:**

The main weaknesses are: assumptions about the defenders, details of the experiments,  the hyperparameter settings and the fairness of comparison in Table 1, and practicality of the proposed method.

**Justification For Why Not Lower Score:**

As agreed by most reviewers, although there are some concerns, the submission has sufficient novelty and should have a good impact.

---

### Decision · Program_Chairs · 2024-01-16

Accept (poster)